# Reweighted Flow Matching via Unbalanced OT for Label-free Long-tailed Generation

## Abstract

Flow matching has recently emerged as a powerful framework for continuous-time generative modeling. However, when applied to long-tailed distributions, standard flow matching suffers from majority bias, producing minority modes with low fidelity and failing to match the true class proportions. In this work, we propose *Unbalanced Optimal Transport Reweighted Flow Matching (UOT-RFM)*, a novel framework for generative modeling under class-imbalanced (long-tailed) distributions that operates without any class label information. Our method constructs the conditional vector field using mini-batch Unbalanced Optimal Transport (UOT) and mitigates majority bias through a principled inverse reweighting strategy. The reweighting relies on a label-free majority score, defined as the density ratio between the target distribution and the UOT marginal. This score quantifies the degree of majority based on the geometric structure of the data, without requiring class labels. By incorporating this score into the training objective, UOT-RFM theoretically recovers the target distribution with first-order correction ($k = 1$) and empirically improves tail-class generation through higher-order corrections ($k > 1$). Our model outperforms existing flow matching baselines on long-tailed benchmarks, while maintaining competitive performance on balanced datasets.

## 1 Introduction

Generative modeling aims to learn a model that can approximate a target data distribution. In recent years, deep generative models have achieved remarkable progress across various domains, such as GANs (Goodfellow et al., 2014; Arjovsky et al., 2017), optimal transport maps (Rout et al., 2022; Fan et al., 2023; Choi et al., 2023), and diffusion models (Ho et al., 2020; Song et al., 2021). Among these, flow matching models (Lipman et al., 2022) have emerged as a promising approach for continuous-time generative modeling. Flow matching learns a continuous normalizing flow (Chen et al., 2018), i.e., a vector field that transports samples from a prior distribution to a target distribution, while avoiding costly numerical likelihood estimation. The flow matching model is trained through regression to a conditional vector field, constructed from the conditional probability path between prior and target samples. Despite these computational advantages, flow matching models face similar challenges as other generative approaches when dealing with real-world data characteristics.

A particularly challenging scenario in real-world data is **long-tailed or imbalanced distributions**, where a few majority (head) classes dominate and many minority (tail) classes are severely underrepresented (Yang et al., 2022). Standard generative models often suffer from **majority bias** under such settings: head classes are well-modeled, but tail classes are badly generated or ignored (Cao et al., 2019; Qin et al., 2023). This leads to several issues, including inaccurate class proportions, reduced sample diversity, and degraded generation quality for tail classes. To mitigate this, various long-tailed generative models have been proposed, such as GAN-based approaches (Rangwani et al., 2021; 2022) and diffusion-based methods (Zhang et al., 2024; Qin et al., 2023). While effective, these methods rely heavily on **explicit class label information** to improve generation for minority classes. Despite these advances, the flow matching framework has not yet been explored in the context of long-tailed generation. In this work, we fill this gap by analyzing how flow matching models behave under class imbalance and proposing a label-free method to improve their performance in long-tailed regimes.

To overcome these challenges, we propose a novel flow matching model based on the Unbalanced Optimal Transport (UOT) (Chizat et al., 2018; Liero et al., 2018), called *UOT-Reweighted Flow*

*Matching (UOT-RFM)*. Our method leverages mini-batch UOT to construct the conditional vector field and mitigates majority bias through a principled *inverse reweighting* scheme with a *label-free majority score*. This score measures class dominance based on the geometric properties of the data and is defined as the density ratio between the target distribution and the UOT marginal. By incorporating this score into the flow matching objective, UOT-RFM adaptively reweights training samples. With first-order correction ($k = 1$), the model recovers the original data distribution. With higher-order corrections ($k > 1$), the model further compensates for majority bias by emphasizing tail samples. This reweighting mechanism enables the model to improve generation quality for underrepresented classes without requiring class labels. Our experiments demonstrate that UOT-RFM significantly outperforms existing flow matching baselines in both tail-class fidelity and accurate recovery of class proportions. Moreover, our model maintains competitive performance on balanced datasets (CIFAR-10 and CIFAR-100). Our contributions can be summarized as follows:

- We propose UOT-RFM, the first flow matching framework for long-tailed generative modeling, built on mini-batch Unbalanced Optimal Transport.
- We introduce a majority score, derived from the density ratio between the target and UOT marginal distributions, enabling sample-wise reweighting without access to class labels.
- We theoretically and empirically show that higher-order correction using the majority score improves tail sample generation while preserving overall performance.
- To the best of our knowledge, UOT-RFM is the first label-free method for long-tailed generative modeling.

## 2 PRELIMINARIES

**Flow Matching** Continuous Normalizing Flows (CNFs) (Chen et al., 2018; Lipman et al., 2022) model the dynamics of the probability densities through a *probability density path* $p : [0, 1] \times \mathbb{R}^d \mapsto \mathbb{R}_{\geq 0}$, where $p_t(\mathbf{x}) := p(t, \mathbf{x})$ denotes the density at time $t$, which transports the initial or source distribution (e.g., Gaussian distribution) $p_0$ to the target data distribution $p_1$. Specifically, the CNF model is defined by the following Ordinary Differential Equation (ODE), governed by a vector field $\mathbf{v} : [0, 1] \times \mathbb{R}^d \mapsto \mathbb{R}^d$, where $\mathbf{v}_t(\mathbf{x}) := \mathbf{v}(t, \mathbf{x})$:

$$\frac{d\mathbf{x}_t}{dt} = \mathbf{v}_t(\mathbf{x}_t), \tag{1}$$

where $\mathbf{x}_t \in \mathbb{R}^d$ denotes the state variable at time $t$, and we use the notation $\mathbf{v}_t(\mathbf{x})$ interchangeably with $\mathbf{v}(t, \mathbf{x})$. Then, the associated flow map $\phi_t(\mathbf{x})$ denotes the solution of this ODE with initial condition $\phi_0(\mathbf{x}) = \mathbf{x}$ and the density at time $t$ is given by $p_t = (\phi_t)_\# p_0$.

Lipman et al. (2022) proposed *flow matching*, a scalable method for training CNFs. The idea is to train the CNF by minimizing a regression loss $\mathcal{L}_{\text{FM}}(\boldsymbol{\theta})$ between the parameterized vector field $\mathbf{v}_t^\theta$ and the ground-truth vector field $\mathbf{u}_t$ that generates the probability path $p_t$. However, a major challenge is that the marginal ground-truth vector field $\mathbf{u}_t$ is intractable.

$$\mathcal{L}_{\text{FM}}(\boldsymbol{\theta}) = \mathbb{E}_{t \sim \mathcal{U}[0,1], \mathbf{x}_t \sim p_t(\mathbf{x}_t)} \|\mathbf{v}_\theta(t, \mathbf{x}_t) - \mathbf{u}_t(\mathbf{x}_t)\|_2^2. \tag{2}$$

To overcome this, the flow matching (Lipman et al., 2022; Tong et al., 2024) introduces a conditional flow matching. Instead of matching $\mathbf{u}_t$, the model is trained to regress the tractable *conditional vector field* $\mathbf{u}_t(\mathbf{x}_t|\mathbf{z})$, which generates a *conditional probability path* $p_t(\mathbf{x}_t|\mathbf{z})$, where $\mathbf{z}$ denotes sample pairs $(\mathbf{x}_0, \mathbf{x}_1)$. The sample pairs $(\mathbf{x}_0, \mathbf{x}_1)$ follow the joint distribution (couplings) of $\pi(\mathbf{z}) = \pi(\mathbf{x}_0, \mathbf{x}_1)$. The training objectives are given by

$$\mathcal{L}_{\text{CFM}}(\boldsymbol{\theta}) = \mathbb{E}_{t \sim \mathcal{U}[0,1], \mathbf{z} \sim \pi(\mathbf{z}), \mathbf{x}_t \sim p_t(\mathbf{x}_t|\mathbf{z})} \|\mathbf{v}_\theta(t, \mathbf{x}_t) - \mathbf{u}_{t|\mathbf{z}}(\mathbf{x}_t|\mathbf{z})\|_2^2. \tag{3}$$

CFM replaces the intractable marginal vector field with a tractable conditional one based on couplings. In particular, the conditional probability path $p_t(\mathbf{x}_t|\mathbf{z})$ and the associated conditional vector field $\mathbf{u}_t(\mathbf{x}_t|\mathbf{z})$ can be defined as follows (Tong et al., 2024):

$$p_t(\mathbf{x}_t \mid \mathbf{z}) = \mathcal{N}\left(\mathbf{x}_t \mid t\mathbf{x}_1 + (1 - t)\mathbf{x}_0 \mid \sigma^2 I\right), \quad \mathbf{u}_t(\mathbf{x}_t \mid \mathbf{z}) = \mathbf{x}_1 - \mathbf{x}_0, \tag{4}$$

where $\sigma > 0$ is a bandwidth hyperparameter. In this case, the marginal probability path and the marginal vector field that generates this path are given by

$$p_t(\mathbf{x}_t) = \int p_t(\mathbf{x}_t \mid \mathbf{z})\pi(\mathbf{z})d\mathbf{z}, \quad \mathbf{u}_t(\mathbf{x}_t) := \mathbb{E}_{\pi(\mathbf{z})}\left[\frac{\mathbf{u}_t(\mathbf{x} \mid \mathbf{z})p_t(\mathbf{x} \mid \mathbf{z})}{p_t(\mathbf{x})}\right] = \mathbb{E}_{p_t(\mathbf{z}|\mathbf{x}_t)}\left[\mathbf{u}_t(\mathbf{x}_t \mid \mathbf{z})\right]. \tag{5}$$

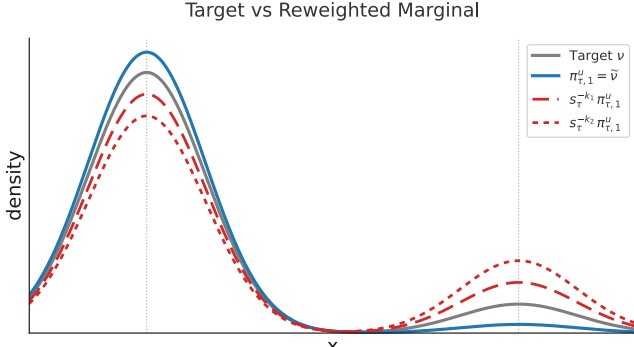

Figure 1: **Comparison of the target data distribution** $\nu$ **and the reweighted marginals** $s_\tau^{-k_1}\pi_{\tau,1}^u$ **from UOT-RFM** with correction order $k$ (where $1 < k_1 < k_2$). The UOT marginal $\pi_{\tau,1}^u$ downweights the minority classes. UOT-RFM adaptively upweights minority modes via the majority score $s_\tau$.

**Initial Coupling in Flow Matching** A key component in training flow matching models is the choice of the initial coupling $\mathbf{z} = (\mathbf{x}_0, \mathbf{x}_1)$ with joint distribution $\pi(\mathbf{z}) = \pi(\mathbf{x}_0, \mathbf{x}_1)$. **The choice of coupling crucially determines the training dynamics of flow matching models**, because the obtained model $\mathbf{v}_\theta(\mathbf{x}_t) \approx \mathbf{u}_t(\mathbf{x}_t)$ relies on aggregating the conditional vector field over paired samples $p_t(\mathbf{z}|\mathbf{x}_t)$ (Eq. 5). The original flow matching framework (Lipman et al., 2022) employs an independent coupling between the source and target distributions, i.e., $\pi(\mathbf{x}_0, \mathbf{x}_1) = \mu(\mathbf{x}_0) \otimes \nu(\mathbf{x}_1)$. However, such independence often leads to curved trajectories that incur high computational costs during sampling (Liu et al., 2023). To improve couplings, recent works adopted the *Optimal Transport (OT)* approaches between mini-batches (Pooladian et al., 2023; Tong et al., 2024). Note that the Kantorovich formulation of the Optimal Transport is given by

$$C_{OT}(\mu, \nu) := \inf_{\pi \in \Pi(\mu,\nu)} \left[ \int_{\mathcal{X} \times \mathcal{Y}} c(\mathbf{x}, \mathbf{y}) d\pi(\mathbf{x}, \mathbf{y}) \right], \tag{6}$$

where $\Pi(\mu, \nu)$ denotes the set of all joint probability measures on $\mathcal{X} \times \mathcal{Y}$ whose marginals are $\mu$ and $\nu$ respectively. Here, the optimal coupling $\pi$ is defined as the minimizer of the transport cost $c(\mathbf{x}, \mathbf{y})$ between empirical measures of mini-batches from the source samples $\mathbf{x}_0$ and target samples $\mathbf{x}_1$.

## 3 METHOD

In this section, we present our model, ***UOT-Reweighted Flow Matching (UOT-RFM)***, that addresses the majority bias of flow matching models on long-tailed distributions. Our model leverages mini-batch UOT coupling, which naturally provides a ***majority score*** for each target sample. Intuitively, we compensate for majority bias by reweighting each target sample with the majority score. In Sec 3.1, we introduce the UOT problem. In Sec 3.2, we introduce our UOT-RFM model.

### 3.1 UNBALANCED OPTIMAL TRANSPORT

Our proposed method builds on the *Unbalanced Optimal Transport (UOT) problem* (Chizat et al., 2018; Liero et al., 2018). In this regard, we introduce the UOT problem and its key properties, which will be leveraged in our approach. In the classical OT problem (Eq. 6), the marginal distributions of the coupling $\pi$ are constrained to *exactly* match the source and target distributions, i.e., $\pi_0 = \mu$ and $\pi_1 = \nu$. Although this exact marginal matching is a core principle of OT, it also makes the formulation highly sensitive to outliers (Balaji et al., 2020; Séjourné et al., 2022; Gazdieva et al., 2025; Choi et al., 2023). In contrast, the *Unbalanced Optimal Transport* formulation relaxes these hard marginal constraints by introducing divergence penalties between the marginals $\pi_0, \pi_1$ and the source/target measures $\mu, \nu$. This relaxation enables approximate transport, thereby improving robustness to outliers. Formally, the Kantorovich-type UOT formulation is given by:

$$C_{UOT}(\mu, \nu) = \inf_{\pi \in \mathcal{M}_+(\mathcal{X} \times \mathcal{Y})} \left[ \int_{\mathcal{X} \times \mathcal{Y}} c(\mathbf{x}, \mathbf{y}) d\pi(\mathbf{x}, \mathbf{y}) + \tau_1 D_{\Psi_1}(\pi_0 \| \mu) + \tau_2 D_{\Psi_2}(\pi_1 \| \nu) \right], \tag{7}$$

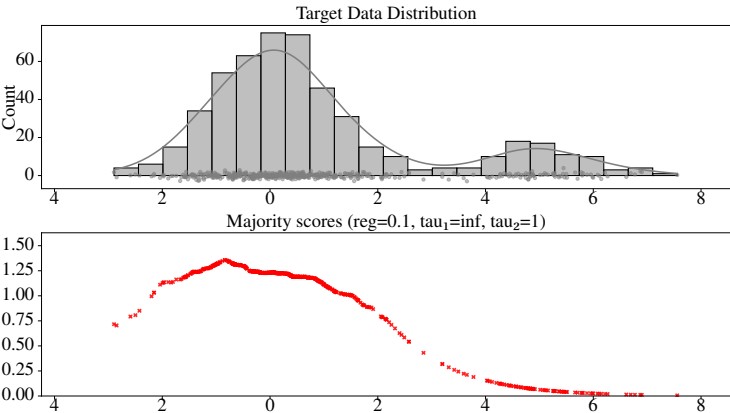

Figure 2: **Example of majority score $s_\tau$ computed via mini-batch UOT.** The source distribution is standard Gaussian $\mathcal{N}(0, I)$, and the target distribution is a Gaussian mixture (top). The majority scores (bottom) are higher in majority regions and lower in minority regions.

where we assume $c(\mathbf{x}, \mathbf{y}) = 1/2\|\mathbf{x} - \mathbf{y}\|_2^2$ and $\tau_1, \tau_2 > 0$ control the strength of the marginal matching penalties. Here, $\mathcal{M}_+(\mathcal{X} \times \mathcal{Y})$ indicates the set of positive Radon measures on $\mathcal{X} \times \mathcal{Y}$. The terms $D_{\Psi_1}(\pi_0\|\mu)$ and $D_{\Psi_2}(\pi_1\|\nu)$ are two $f$-divergences that penalize deviations of the coupling marginals $\pi_0, \pi_1$ from the source $\mu$ and target $\nu$, respectively. The $f$-divergence $D_\Psi$ is defined as $D_\Psi(\pi_i\|\eta) = \int \Psi\left(\frac{d\pi_i(\mathbf{x})}{d\eta(\mathbf{x})}\right) d\eta(\mathbf{x})$ for the convex function $\Psi$. This relaxed formulation allows the optimal UOT coupling $\pi^u$ (which depends on $\tau_1$ and $\tau_2$) to softly match the marginals, i.e., $\pi_0^u \approx \mu$ and $\pi_1^u \approx \nu$, in contrast to the exact marginal constraints in standard OT.

Moreover, the UOT problem can represent exact matching of one marginal by appropriately setting the divergence penalty. Specifically, if $\Psi_i$ is the convex indicator function $\iota$ at $\{1\}$, then $D_\iota(\pi_i\|\eta) = 0$ if $\pi_i = \eta$ a.s., and $\infty$ otherwise. For example, setting $\Psi_1 = \iota$ (i.e., $\tau_1 = \infty$) yields the **source-fixed UOT problem**, where $\pi_0^u = \mu$ and $\pi_1^u \approx \nu$. In this case, the optimal coupling depends only on the single parameter $\tau$. In our approach, we employ this **source-fixed UOT** formulation to ensure that the initial distribution of the flow matching model aligns exactly with the source distribution.

## 3.2 PROPOSED METHOD

**Problem Statement** Our goal is to develop a generative model that performs well on **long-tailed data distributions, without relying on class labels**. Formally, we are given a long-tailed dataset $\mathcal{D} = \{(\mathbf{x}^i, \mathbf{y}^i)\}_{i=1}^N$, where each $\mathbf{x}_i$ is an input image and $\mathbf{y}^i \in \mathcal{C}$ is its corresponding class label. In the long-tailed setting, a small number of classes (**head classes**) dominate with many samples, while most classes (**tail classes**) have only a few, resulting in severe class imbalance. Specifically, let $\mathcal{C} = \{c_1, c_2, \ldots, c_M\}$ denote the set of $M$ classes, ordered by decreasing sample count such that $n_1 \geq n_2 \geq \cdots \geq n_M$, where $n_j$ is the number of training samples in class $c_j$. The imbalance ratio is defined as $\mathcal{I} = n_M/n_1$. For instance, under an exponentially decaying class distribution, the class sizes follow $n_i = \lfloor n_1 \cdot \mathcal{I}^{\frac{i}{M-1}} \rfloor$. The objective of long-tailed generative modeling is to learn a model that can faithfully generate all classes, including the tail classes. Moreover, in our setting, the model is trained and evaluated *without using any class label information*.

**Majority Score** Our method leverages the **mini-batch UOT coupling** $\pi^u$ and the induced **majority score**, defined as the density ratio $s_\tau$. This score is used to address the majority bias of flow matching models on long-tailed distributions by inversely reweighting each target sample during training. Intuitively, the optimal UOT coupling $\pi^u$ favors transport plans where a small increase in the divergence penalty $D_\Psi$ leads to a large decrease in the transport cost $c(\mathbf{x}, \mathbf{y})$ (Eq. 7). Consequently, $\pi^u$ tends to concentrate mass on high-density (majority) modes while down-weighting low-density (tail or outlier) modes, which contribute little mass and incur high transport costs (Fig. 2). This mechanism underlies the robustness of UOT to outliers, as it effectively mitigates their influence (Balaji et al., 2020; Séjourné et al., 2022; Choi et al., 2023).

Based on this property, we formally define the ***majority score*** under the source-fixed UOT problem as

$$s_\tau(\mathbf{x}_1) := \frac{d\pi_{\tau,1}^u}{d\nu}(\mathbf{x}_1) > 0, \tag{8}$$

where $\pi_{\tau,1}^u$ denotes the target marginal of the UOT coupling, and $s_\tau : \mathbb{R}^d \to \mathbb{R}_{>0}$ is defined on the target distribution space. Since we employ the source-fixed UOT formulation $\tau_1 = \infty$, the coupling depends only on $\tau_2$. Thus, we simply denote $\tau = \tau_2$ for clarity. Intuitively, the majority score measures how strongly each target sample is emphasized by the UOT coupling. $s_\tau > 1$ indicates emphasized majority samples, while $s_\tau \ll 1$ correspond to down-weighted outlier samples. Here, note that this process is entirely **unsupervised**, i.e., label-free. The down-weighting of outlier modes is conducted by the intrinsic geometric structure of probability distributions (See Appendix B for the formal theoretical relationship between the marginal distributions of $\pi_\tau^u$ and the original source and target distributions).

**Proposed Method**  We now introduce our learning objective for modeling long-tailed distributions. Our method consists of two key components: (1) mini-batch (source-fixed) UOT coupling $\pi_\tau^u$ and (2) rebalancing minority samples through importance (over-)correction using majority score $s_\tau(\cdot)$. Our corrected conditional flow matching objective with correction order $k \geq 1$ is defined as follows (Algorithm 1):

$$\mathcal{L}_{\text{ours},k}(\boldsymbol{\theta}) = \mathbb{E}_{t\sim\mathcal{U}[0,1],\mathbf{z}\sim\pi_\tau^u(\mathbf{z}),\mathbf{x}_t\sim p_t(\mathbf{x}_t|\mathbf{z})} \left[ s_\tau(\mathbf{x}_1)^{-k}\|\mathbf{v}_\theta(t,\mathbf{x}_t) - \mathbf{u}_{t|\mathbf{z}}(\mathbf{x}_t|\mathbf{z})\|_2^2 \right], \tag{9}$$

where the conditioning variable $\mathbf{z} = (\mathbf{x}_0, \mathbf{x}_1)$. Compared with standard flow matching (Eq. 4), our formulation employs the UOT coupling $\pi^u$ for pairing $\mathbf{z}$ and introduce an additional weighting factor $s_\tau(\mathbf{x}_1)^{-k}$ that rebalances majority and minority samples. Our method is motivated by the following bias correction theorem (see Appendix C for proof):

**Theorem 3.1.** *Let $\pi_\tau^u$ be the optimal source-fixed UOT coupling between $\mu$ and $\nu$ with $\tau_2 = \tau > 0$ and assume that its target marginal satisfies $\nu \ll \pi_{\tau,1}^u$, i.e., $\nu$ is absolutely continuous w.r.t. $\pi_{\tau,1}^u$. Training a flow matching model with $\pi_\tau^u$ yields the biased distribution $p_1 = \pi_{\tau,1}^u \neq \nu$. However, applying the first-order correction (our method with $k = 1$) recovers the true target distribution $\nu$.*

$$\mathcal{L}_{\text{ours,k=1}}(\boldsymbol{\theta}) = \mathbb{E}_{t\sim\mathcal{U}[0,1],\mathbf{z}\sim\pi_\tau^u(\mathbf{z}),\mathbf{x}_t\sim p_t(\mathbf{x}_t|\mathbf{z})} \left[ s_\tau(\mathbf{x}_1)^{-1}\|\mathbf{v}_\theta(t,\mathbf{x}_t) - \mathbf{u}_{t|\mathbf{z}}(\mathbf{x}_t|\mathbf{z})\|_2^2 \right]. \tag{10}$$

*More generally, UOR-RFM with correction order $k$ generates $p_1 \propto s_\tau^{-k}\pi_{\tau,1}^u = s_\tau^{-(k-1)}\nu$.*

The assumption $\nu \ll \pi_{\tau,1}^u$ ensures that all target modes, including tail classes, retain nonzero density under the UOT marginal, thereby enabling correction with $s_\tau(\mathbf{x}_1)^{-k}$. Moreover, we impose a **source-fixed** condition on the UOT coupling (i.e., $\pi_0^u = \mu$) to ensure that the initial distribution of the flow matching model remains aligned with the source distribution.

Theorem 3.1 shows that training a flow matching model with UOT coupling (UOT-CFM, (Eyring et al., 2024)) yields a biased generated distribution $p_1 = \pi_{\tau,1}^u \neq \nu$. In particular, the distribution $\pi_{\tau,1}^u$ magnifies the majority modes while suppressing the tail modes. This bias can be corrected by applying inverse weighting with the majority score $s_\tau$. Building on this, our method further addresses the majority bias of standard flow matching models by **over-correction** ($k > 1$) with the majority score. Intuitively, this over-correction amplifies the contribution of tail-class samples with $s_\tau(\cdot) < 1$. In contrast to OT-CFM (Tong et al., 2024), which adopts mini-batch OT coupling, our UOT-based approach provides an **unsupervised estimate of the majority score without requiring additional information such as tail-class label**. Refer to Appendix D for a discussion of related works.

**Minibatch UOT Approximation**  Following mini-batch OT approaches (Pooladian et al., 2023; Tong et al., 2024), we approximate the UOT coupling $\pi_\tau^u$ using a mini-batch formulation similar to (Fatras et al., 2021). In practice, we adopt the POT library (Flamary et al., 2021) to compute mini-batch UOT with entropic regularization (Chizat et al., 2016; Frogner et al., 2015). Specifically, for each mini-batch of training data $(\{\mathbf{x}_0^i\}_{i=1}^B, \{\mathbf{x}_1^j\}_{j=1}^B)$, the mini-batch coupling $\hat{\pi}_\tau^u$ is computed between empirical measures $\hat{\mu} = \frac{1}{|B|}\sum_i \delta_{\mathbf{x}_0^i}$ and $\hat{\nu} = \frac{1}{|B|}\sum_j \delta_{\mathbf{x}_1^j}$. Based on this, the majority score is estimated by the probability mass ratio:

$$\hat{s}_\tau(\mathbf{x}_1^j) := \frac{\hat{\pi}_{\tau,1}^u}{\hat{\nu}}(\mathbf{x}_1^j) = |B|\hat{\pi}_{\tau,1}^u(\mathbf{x}_1^j). \tag{11}$$

Table 1: **Evaluation of marginal distribution matching under the LT→LT setting for the long-tailed benchmarks.** We report FID scores (↓) on CIFAR-10-LT and CIFAR-100-LT with two imbalance ratios: $\mathcal{I} = 0.01$ and $\mathcal{I} = 0.001$.

| Model | CIFAR-10-LT | | CIFAR-100-LT | |
|---|---|---|---|---|
| | $\mathcal{I} = 0.01$ | $\mathcal{I} = 0.001$ | $\mathcal{I} = 0.01$ | $\mathcal{I} = 0.001$ |
| I-CFM | 14.57 | 17.54 | 25.55 | 31.86 |
| OT-CFM | 17.31 | 21.26 | 31.34 | 38.37 |
| UOT-CFM | 14.25 | 18.13 | 25.33 | 31.83 |
| ours | **11.03** | **12.84** | **15.37** | **18.40** |

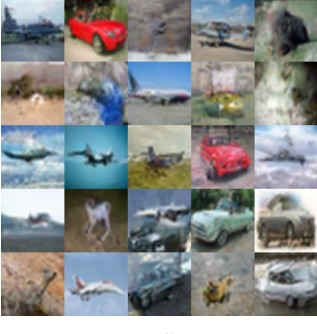
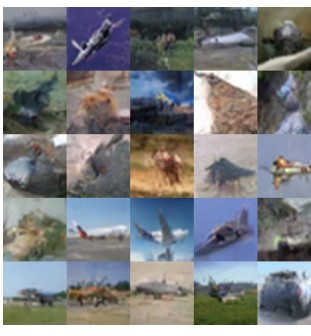
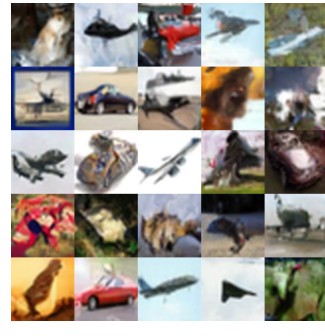

|  (a) I-CFM  |  (b) OT-CFM  |  (c) UOT-RFM  |
|---|---|---|

Figure 3: **Qualitative comparison of generated samples** from flow matching models trained on CIFAR-10-LT with imbalance ratio $\mathcal{I} = 0.01$. UOT-RFM produces more diverse images compared to other baselines.

## 4    EXPERIMENTS

In this section, we evaluate our model from the following perspectives.

- In Sec 4.1, we evaluate our model on the long-tailed image datasets and analyze the majority bias of flow matching models.
- In Sec 4.2, we assess our model on the standard balanced image datasets, showing that our model remains competitive with small-order correction.
- In Sec 4.3, we conduct ablation studies to investigate the effects of the correction order $k$ and the marginal matching parameter $\tau$.

In each experiment, our model is compared with several flow matching baselines: independent coupling (*I-CFM*, (Lipman et al., 2022; Tong et al., 2024)), OT coupling (*OT-CFM*, (Tong et al., 2024; Pooladian et al., 2023)), and *UOT coupling* (UOT-CFM, (Eyring et al., 2024)). Implementation details are provided in Appendix E.

### 4.1    EVALUATION ON LONG-TAILED GENERATION

We evaluate our model on two long-tailed generation benchmarks: **CIFAR-10-LT** and **CIFAR-100-LT** (Cao et al., 2019). These datasets are constructed by subsampling the balanced CIFAR-10 and CIFAR-100 datasets (Krizhevsky et al., 2009) according to an exponential decaying long-tailed class distribution. The degree of imbalance is quantified by the imbalance ratio $\mathcal{I}$, defined as the ratio between the sample sizes of the most and least frequent classes, i.e., $\mathcal{I} = \min_i\{n_i\}/\max_i\{n_i\}$.

**Long-Tailed Generation**    We first assess whether our model accurately approximates the true marginal distribution under long-tailed settings. To this end, we consider two evaluation settings, while keeping the training data fixed to the long-tailed dataset:

- **LT→LT**: The test set is also long-tailed (e.g., CIFAR-10-LT test set). This setting assesses how well a model captures the long-tailed distribution.

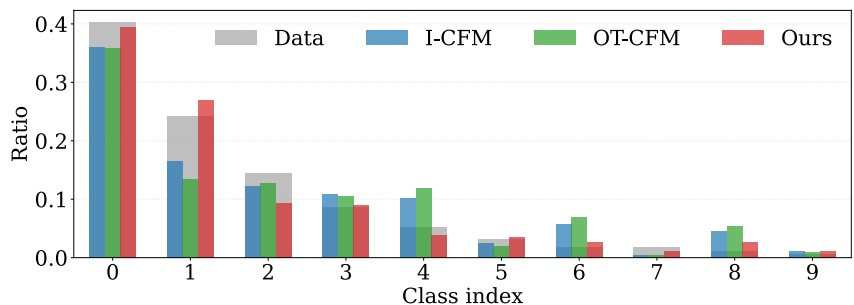

Figure 4: **Generated class distribution on CIFAR-10-LT** with $\mathcal{I} = 0.01$. The average Normalized Class Ratio Errors (NCREs) are: I-CFM = 0.84, OT-CFM = 1.02, and ours = 0.40.

Table 3: **Quantitative evaluation of mode coverage** under the LT→LT setting with $\mathcal{I} = 0.01$.

| Model | CIFAR-10-LT | | | CIFAR-100-LT | | |
|---|---|---|---|---|---|---|
| | Precision (↑) | Recall (↑) | F1 (↑) | Precision (↑) | Recall (↑) | F1 (↑) |
| I-CFM | 0.71 | 0.29 | 0.41 | 0.62 | 0.29 | 0.40 |
| OT-CFM | **0.73** | 0.24 | 0.36 | **0.73** | 0.24 | 0.36 |
| UOT-CFM | **0.73** | 0.29 | 0.42 | **0.73** | 0.27 | 0.40 |
| ours | 0.62 | **0.41** | **0.49** | 0.69 | **0.32** | **0.44** |

- **LT→Balanced**: The test set is the original balanced dataset (e.g., CIFAR-10 test set). This setting examines whether a model trained on imbalanced data can recover a balanced distribution. This evaluation setup is often used in supervised long-tailed learning when class labels are available (Zhang et al., 2024; Qin et al., 2023).

Our primary evaluation protocol is LT→LT, because our goal is to directly model and approximate the long-tailed data distribution without relying on class supervision. For completeness, we also report LT→Balanced results under an imbalance ratio of $\mathcal{I} = 0.01$.

Table 1 provides FID (Heusel et al., 2017) scores on CIFAR-10 and CIFAR-100 under the **LT→LT setting** for two class imbalanced ratios, i.e., $\mathcal{I} \in \{0.01, 0.001\}$. Fig. 3 shows qualitative comparisons of generated samples from baseline flow matching models trained on CIFAR-10-LT with $\mathcal{I} = 0.01$. In both imbalance ratios, UOT-RFM achieves significant improvement in the FID score, demonstrating a more accurate approximation of the long-tailed data distribution. Note that this performance gain comes with minimal computational overhead: UOT-RFM requires only about 7% more training time compared to OT-CFM.

Moreover, Table 2 reports FID scores under the **LT→Balanced** setting with $\mathcal{I} = 0.01$. The trends are consistent with those in the LT→LT evaluation. UOT-RFM outperforms all baselines, achieving the lowest FID score—particularly in the more challenging CIFAR-100-LT case. We omit results for the more extreme case of $\mathcal{I} = 0.001$ in this setting. Note that $\mathcal{I} = 0.01$ already represents a highly challenging scenario under a label-free setting. This requires the model to counterbalance a 100:1 disparity between the most and least frequent classes, without access to any class label information.

Table 2: **Evaluation of marginal distribution matching under the LT→Balanced setting for the long-tailed benchmarks.** We report FID scores (↓) with imbalance ratio $\mathcal{I} = 0.01$.

| Model | CIFAR-10-LT | CIFAR-100-LT |
|---|---|---|
| I-CFM | 25.46 | 24.39 |
| OT-CFM | 27.51 | 29.19 |
| UOT-CFM | 24.94 | 24.05 |
| ours | **24.06** | **16.83** |

**Addressing Majority Bias** We further investigate how well each model handles majority bias through a detailed class-wise evaluation.

First, we **compare the class distribution of generated samples to that of the ground-truth test set**. Since our model is *unconditional* (i.e., it does not take class labels as input), we utilize

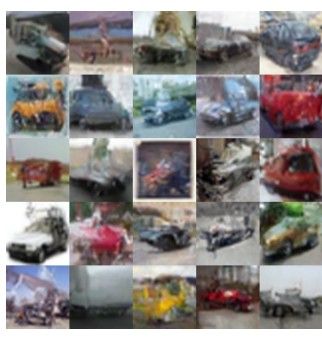 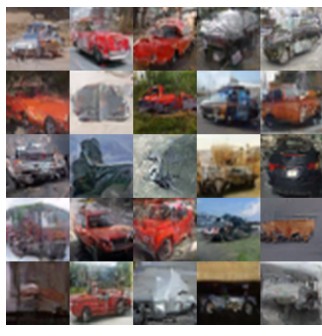 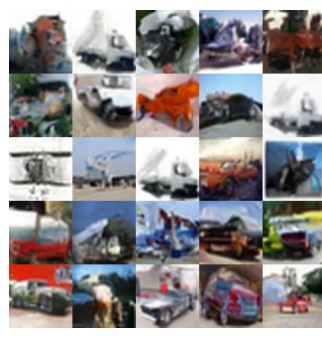

|  (a) I-CFM  |  (b) OT-CFM  |  (c) UOT-RFM  |

Figure 5: **Qualitative comparison of generated *tail* samples** from flow matching models trained on CIFAR-10-LT with $\mathcal{I} = 0.01$. Samples with the highest confidence scores (as predicted by a pretrained classifier) are visualized.

pretrained classifiers on CIFAR-10 and CIFAR-100 to assign proxy labels to generated samples[1]. Fig. 4 shows the generated class distribution on CIFAR-10-LT (see Fig. 7 in Appendix for CIFAR-100-LT). Compared to I-CFM and OT-CFM, our UOT-RFM produces a class distribution that more closely matches the ground-truth long-tailed distribution. To quantify this, we compute the **Normalized Class Ratio Error (NCRE)**, defined as the relative deviation between the generated and true class proportions:

$$\text{NCRE}_i = \frac{\left|r_{gen,i} - r_{data,i}\right|}{r_{data,i}}, \tag{12}$$

where $r_{gen,i}$ denotes the proportion of generated images assigned to class $c_i$ (via proxy labels) and $r_{data,i}$ is the ground-truth class proportion. Since this metric is normalized by the true proportion, misalignment in tail classes is penalized more heavily. The class-average NCRE scores are 0.84 for I-CFM, 1.02 for OT-CFM, and 0.40 for UOT-RFM (see Figs 8 and 9 for classwise scores on CIFAR-10-LT and CIFAR-100-LT). These results demonstrate that UOT-RFM most accurately approximates the target class distribution.

Second, we evaluate the models using Precision, Recall, and F1-score (Kynkäänniemi et al., 2019), which **provide explicit measurements of mode coverage and balance in sample generation**. Table 3 presents the scores of each flow matching model on CIFAR-10-LT and CIFAR-100-LT with imbalance ratio $\mathcal{I} = 0.01$. Across both datasets, UOT-RFM achieves the highest Recall, demonstrating superior coverage of tail modes. While OT-CFM obtains the highest precision metric, UOT-RFM achieves the best F1-score, reflecting a more balanced trade-off between Precision and Recall.

Finally, we assess the **fidelity of generated samples from tail classes**. Fig. 5 shows representative generated images, selected by the highest-confidence predictions of the pretrained classifier. The generated samples from I-CFM and OT-CFM often exhibit noisy artifacts, whereas UOT-RFM produces cleaner, higher-quality images without such artifacts. In addition, we evaluate classwise negative log-likelihood (NLL ($\downarrow$)) to further assess distributional fidelity. Due to space constraints, full results are provided in Appendix (Figs. 10 and 12). UOT-RFM consistently achieves lower NLL across almost all classes compared to I-CFM and OT-CFM. For example, the mean NLL on CIFAR-10-LT is 3.88 for UOT-RFM, compared to 4.02 and 4.06 for I-CFM and OT-CFM, respectively.

In summary, these experimental results demonstrate that UOT-RFM more effectively addresses the majority bias than existing flow matching models—achieving better alignment with the target class distribution, improved minority class coverage, and higher-fidelity sample generation from tail modes.

### 4.2 EVALUATION ON STANDARD BALANCED GENERATION

To further test the general applicability of our method beyond long-tailed distributions, we evaluate UOT-RFM on the standard balanced benchmarks: CIFAR-10 and CIFAR-100. Table 4 shows the results. As suggested in Theorem 3.1, UOT-RFM with exact correction ($k = 1$) can be applied to balanced generative modeling. Our model achieves performance comparable to existing flow

---

[1]https://github.com/chenyaofo/pytorch-cifar-models

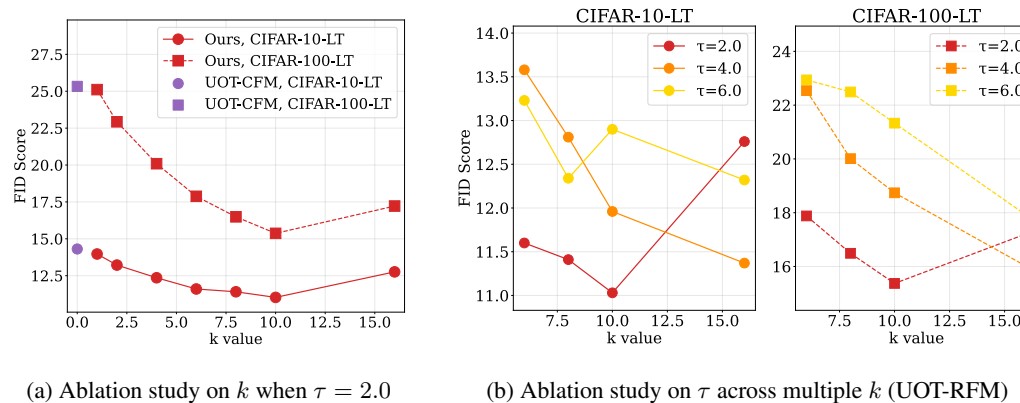

(a) Ablation study on $k$ when $\tau = 2.0$       (b) Ablation study on $\tau$ across multiple $k$ (UOT-RFM)

Figure 6: **Ablation studies** on the correction order $k$ and the marginal matching strength $\tau$.

matching baselines. This result demonstrates that UOT-RFM is not limited to long-tailed settings but also remains competitive on standard balanced distributions.

### 4.3 ABLATION STUDY

**Correction Order** $k$   We conduct an ablation study on the correction order $k$ to examine its impact on performance. Fig. 6a shows that introducing correction consistently improves FID scores over UOT-CFM (i.e., UOT-RFM without correction). Interestingly, the best FID scores are achieved at $k = 10 \gg 1$, rather than with the exact correction $k = 1$. We attribute this to the need for stronger upweighting of minority samples in practice, in order to compensate for

Table 4: **FID scores ($\downarrow$) on standard balanced benchmarks.**

| Model | CIFAR-10 | CIFAR-100 |
|---|---|---|
| I-CFM | 3.78 | 6.39 |
| OT-CFM | 3.64 | **6.14** |
| UOT-CFM | 3.62 | 6.45 |
| ours | **3.58** | 6.54 |

the exponentially decaying class distribution under long-tailed settings. Overall, UOT-RFM is robust to correction order $k$ and consistently outperforms other baseline models for all $1 \le k \le 16$.

**Marginal Matching Intensity** $\tau$   We perform an ablation study on the marginal matching intensity $\tau = \tau_2$ in the mini-batch UOT (Eq. 7). The parameter $\tau$ controls the degree to penalize the marginal errors in the UOT problem. Hence, increasing $\tau$ leads to the more closely matched UOT marginal $\pi_1^u$. Consequently, the majority score becomes close to one, $s_\tau(y) \approx 1$, with smaller variance between majority and minority samples. Fig. 6b shows how the FID scores change according to the different values of $\tau$. Therefore, when $\tau$ is small, even a modest correction order $k$ is sufficient to strongly adjust the generated distribution $p_1 \propto s_\tau^{-(k-1)} \nu$ (Thm. 3.1). We observe that larger $\tau$ values reduce the sensitivity of UOT-RFM to the correction order $k$, stabilizing performance across different settings. However, the best FID scores for long-tailed generation are achieved at a smaller $\tau$, where the model can better adaptively upweight tail-class samples.

## 5 CONCLUSION

In this paper, we introduced UOT-Reweighted Flow Matching (UOT-RFM), a flow matching model for long-tailed distributions. By leveraging Unbalanced Optimal Transport, we introduced a label-free majority score to correct majority bias through inverse weighting and higher-order corrections. We theoretically justified our reweighting scheme and demonstrated its practical effectiveness across long-tailed benchmarks, where UOT-RFM achieves superior tail-class fidelity, balanced sample generation, and state-of-the-art performance among flow matching models. A limitation of UOT-RFM is that it requires training the model from scratch with the reweighting scheme. In contrast, test-time guidance methods operate on pretrained models and do not require retraining. Moreover, they can be orthogonally combined with UOT-RFM. Exploring how our approach can be extended to test-time controllable generation would be an interesting direction for future work.

ETHICS STATEMENT

This work introduces UOT-Reweighted Flow Matching (UOT-RFM), a generative modeling framework aimed at mitigating majority bias in flow matching methods when trained on long-tailed data distributions. By more accurately capturing underrepresented (tail) classes, our approach contributes to fairness in generative modeling. All experiments are conducted on publicly available benchmark datasets (CIFAR-10/100 and their long-tailed variants), which do not include sensitive or personal information. Our study does not involve human subjects or raise privacy, security, or legal concerns, and adheres to established standards of research integrity.

REPRODUCIBILITY STATEMENT

To ensure reproducibility, we provide the implementation code in the supplementary material. Detailed descriptions of the training setup, architecture, and hyperparameters are included in Appendix E. The derivation and complete proof of Theorem 3.1 can be found in Appendix C. All datasets used in our experiments (CIFAR-10/100 and their long-tailed variants) are publicly available.

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

---

**Algorithm 1** Minibatch UOT-Reweighted Flow Matching (UOT-RFM)

---

**Input:** Empirical or sampleable distributions $\mu, \nu$, bandwidth $\sigma$, batch size $b$, initial network $\mathbf{v}_\theta$, sinkhorn target marginal weight $\tau$, weight power scale $k$.
**Initialize:** $\tau_1 \leftarrow \infty$ // Source-fixed UOT
**while** Training **do**
  Sample batches of size $b$ *i.i.d.* from the datasets:
  $\mathbf{x}_0 \sim \mu; \ \mathbf{x}_1 \sim \nu$
  $\pi_\tau^u \leftarrow \text{UOT}(\mathbf{x}_1, \mathbf{x}_0, \tau)$ // Source-fixed UOT $\tau_1 = \infty, \tau_2 = \tau$
  $(\mathbf{x}_0, \mathbf{x}_1) \sim \pi_\tau^u; \ t \sim \mathcal{U}(0,1)$
  $\boldsymbol{\mu}_t \leftarrow t\mathbf{x}_1 + (1-t)\mathbf{x}_0$
  $\mathbf{x} \sim \mathcal{N}(\mu_t, \sigma^2 I)$
  Calculate $\hat{s}_\tau(\mathbf{x}_1)$ from Equation (11)
  $\mathcal{L}_{ours}(\theta) \leftarrow \hat{s}_\tau(\mathbf{x}_1)^{-k} \|\mathbf{v}_\theta(t, \mathbf{x}) - (\mathbf{x}_1 - \mathbf{x}_0)\|^2$
  $\theta \leftarrow \text{Update}(\theta, \nabla_\theta \mathcal{L}_{ours}(\theta))$
**end while**
**return** $\mathbf{v}_\theta$

---

## A  USE OF LARGE LANGUAGE MODELS

We acknowledge the use of a large language model (OpenAI ChatGPT) as a general-purpose writing assistant in the preparation of this work. Specifically, the LLM was used to improve grammar, phrasing, and clarity of the text. The authors take full responsibility for all scientific content presented in this paper.

## B  UNBALANCED OPTIMAL TRANSPORT

Recall that the classical OT problem assumes an exact transport between two distributions $\mu$ and $\nu$, i.e., $\pi_0 = \mu, \pi_1 = \nu$. This exact matching constraint makes OT sensitive to outliers (Balaji et al., 2020; Séjourné et al., 2022) and vulnerable to class imbalance (Eyring et al., 2024). To address these issues, the *Unbalanced Optimal Transport (UOT)* problem (Chizat et al., 2018; Liero et al., 2018) relaxes the hard marginal constraints by introducing divergence penalties with regularization parameters $\tau_1, \tau_2 > 0$:

$$C_{UOT}(\mu, \nu) = \inf_{\pi \in \mathcal{M}_+(\mathcal{X} \times \mathcal{Y})} \left[ \int_{\mathcal{X} \times \mathcal{Y}} c(\mathbf{x}, \mathbf{y}) \, d\pi(\mathbf{x}, \mathbf{y}) + \tau_1 D_{\Psi_1}(\pi_0 \| \mu) + \tau_2 D_{\Psi_2}(\pi_1 \| \nu) \right], \quad (13)$$

where $\mathcal{M}_+(\mathcal{X} \times \mathcal{Y})$ denotes the set of nonnegative finite measures on $\mathcal{X} \times \mathcal{Y}$. Here, $D_{\Psi_1}$ and $D_{\Psi_2}$ are $f$-divergences generated by convex functions $\Psi_i$, penalizing discrepancies between the marginals $\pi_0, \pi_1$ and $\mu, \nu$, respectively. Hence, in the UOT problem, the marginals are only *approximately matched* to $\mu$ and $\nu$, in the sense that $\pi_0$ and $\pi_1$ are close to $\mu$ and $\nu$ with respect to the divergence penalties, rather than being exactly equal as in OT. Intuitively, UOT can be viewed as solving OT between divergence-relaxed marginals $\pi_0, \pi_1$ and the target measures $\mu, \nu$ (Choi et al., 2023). This relaxation provides robustness to outliers (Balaji et al., 2020) and improved adaptability to class imbalance between $\mu$ and $\nu$ (Eyring et al., 2024).

Similar to the standard OT problem, the UOT problem also admits a *dual formulation* (Choi et al., 2023; Gallouët et al., 2021; Vacher & Vialard, 2023):

$$C_{UOT}(\mu, \nu) = \sup_{u(\mathbf{x}) + v(\mathbf{y}) \le c(\mathbf{x}, \mathbf{y})} \left[ \int_{\mathcal{X}} -\tau_1 \Psi_1^* \left( -\tfrac{1}{\tau_1} u(\mathbf{x}) \right) \, d\mu(\mathbf{x}) + \int_{\mathcal{Y}} -\tau_2 \Psi_2^* \left( -\tfrac{1}{\tau_2} v(\mathbf{y}) \right) \, d\nu(\mathbf{y}) \right],$$
$$(14)$$

where $u$ and $v$ are continuous functions on $\mathcal{X}$ and $\mathcal{Y}$. Here, $f^*$ denotes the *convex conjugate* of $f$, i.e., $f^*(z) = \sup_{t \in \mathbb{R}} \{tz - f(t)\}$ for $f : \mathbb{R} \to [-\infty, \infty]$.

This dual problem can be further simplified into a *semi-dual* formulation by eliminating $u$ via the optimality condition:

$$C_{UOT}(\mu, \nu) = \sup_{v \in \mathcal{C}(\mathcal{Y})} \left[ \int_{\mathcal{X}} -\tau_1 \Psi_1^* \left( -\tfrac{1}{\tau_1} v^c(\mathbf{x}) \right) \, d\mu(\mathbf{x}) + \int_{\mathcal{Y}} -\tau_2 \Psi_2^* \left( -\tfrac{1}{\tau_2} v(\mathbf{y}) \right) \, d\nu(\mathbf{y}) \right], \quad (15)$$

where the $c$-transform of $v$ is defined as

$$v^c(\mathbf{x}) = \inf_{\mathbf{y} \in \mathcal{Y}} \big( c(\mathbf{x}, \mathbf{y}) - v(\mathbf{y}) \big).$$

Here, $v^c$ corresponds to the optimal potential $u$ given $v$.

Finally, the relationship between the marginals of the optimal UOT plan $\pi^u$ and the original source and target distributions can be expressed using the optimal UOT potential $v$ from the semi-dual problem:

**Theorem B.1** ((Choi et al., 2023; Gallouët et al., 2021; Vacher & Vialard, 2023)). *Let $v$ be a solution of the dual formulation of the UOT problem between the source distribution $\mu$ and the target distribution $\nu$. Then, the marginal distributions of the optimal UOT plan $\pi^u$ satisfy*

$$d\pi_0^u(\mathbf{x}) = (\Psi_1^*)'\left(-\tfrac{1}{\tau_1} v^c(\mathbf{x})\right) d\mu(\mathbf{x}) \quad and \quad d\pi_1^u(\mathbf{y}) = (\Psi_2^*)'\left(-\tfrac{1}{\tau_2} v(\mathbf{y})\right) d\nu(\mathbf{y}). \tag{16}$$

## C  PROOFS OF THEOREM

In this section, we provide the proof of our bias correction theorem (Theorem 3.1) from the main text. Our proof builds on three key lemmas for the standard flow matching model, originally established in Tong et al. (2024); Lipman et al. (2022), which we restate here for completeness.

**Lemma C.1** (Tong et al. (2024), Theorem 3.1). *The marginal vector field $\mathbf{u}_t$ generates the probability path $p_t(\mathbf{x}_t)$ from initial conditions $p_0(\mathbf{x}_0)$.*

$$p_t(\mathbf{x}_t) = \int p_t(\mathbf{x}_t \mid \mathbf{z}) \pi(\mathbf{z}) d\mathbf{z}, \quad \mathbf{u}_t(\mathbf{x}_t) := \mathbb{E}_{\pi(\mathbf{z})}\left[\frac{\mathbf{u}_t(\mathbf{x} \mid \mathbf{z}) p_t(\mathbf{x} \mid \mathbf{z})}{p_t(\mathbf{x})}\right] = \mathbb{E}_{p_t(\mathbf{z}|\mathbf{x}_t)}\left[\mathbf{u}_t(\mathbf{x}_t \mid \mathbf{z})\right] \tag{17}$$

**Lemma C.2** (Tong et al. (2024), Theorem 3.2). *If $p_t(\mathbf{x}_t) > 0$ for all $\mathbf{x}_t \in \mathbb{R}^d$ and $t \in [0, 1]$, then, up to a constant independent of $\theta$, $\mathcal{L}_{\mathrm{CFM}}$ (Eq. 3) and $\mathcal{L}_{\mathrm{FM}}$ (Eq.2) are equal, and hence*

$$\nabla_\theta \mathcal{L}_{\mathrm{FM}}(\theta) = \nabla_\theta \mathcal{L}_{\mathrm{CFM}}(\theta). \tag{18}$$

**Lemma C.3** (Tong et al. (2024), Proposition 3.4). *Let the initial sample coupling be $\pi(\mathbf{z}_0, \mathbf{z}_1)$ and define the conditional vector probability path and vector field as in Eq. 4. Then, the corresponding marginal probability path $p_t(\mathbf{x}_t)$ satisfies the boundary conditions $p_0 = \pi_0 * \mathcal{N}(\mathbf{x}|0, \sigma^2 I)$ and $p_1 = \pi_1 * \mathcal{N}(\mathbf{x}|0, \sigma^2 I)$, where $*$ denotes the convolution operator. Furthermore, assuming regularity properties of $q_0, q_1$, and the optimal transport plan $\pi$, as $\sigma^2 \to 0$, the marginal path $p_t$ and field $u_t$ minimize (7), i.e., $\mathbf{u}_t$ solves the dynamic optimal transport problem between $\pi_0$ and $\pi_1$. Specifically, $p_0 \to \pi_0$ and $p_1 \to \pi_1$ as $\sigma \to 0$.*

Here, we provide a formal statement of Theorem 3.1 and provide its proof.

**Theorem C.4** (Theorem 3.1). *Let $\pi_\tau^u$ be the optimal source-fixed UOT coupling between $\mu$ and $\nu$ with $\tau_2 = \tau > 0$ and assume that its target marginal satisfies $\nu \ll \pi_\tau^u$. Training a flow matching model with $\pi_\tau^u$ yields the biased distribution $p_1 = \pi_1^u \neq \nu$ Eyring et al. (2024). However, applying the first-order correction (our method with $k = 1$) recovers the true target distribution $\nu$.*

$$\mathcal{L}_{\mathrm{ours},k=1}(\boldsymbol{\theta}) = \mathbb{E}_{t \sim \mathcal{U}[0,1], \mathbf{z} \sim \pi_\tau^u(\mathbf{z}), \mathbf{x}_t \sim p_t(\mathbf{x}_t|\mathbf{z})} \left[ s_\tau(\mathbf{x}_1)^{-1} \|\mathbf{v}_\theta(t, \mathbf{x}_t) - \mathbf{u}_{t|\mathbf{z}}(\mathbf{x}_t|\mathbf{z})\|_2^2 \right]. \tag{19}$$

*where the majority score $s_\tau(\cdot)$ is defined as $s_\tau(\cdot) := \frac{d\pi_1^u}{d\nu}(\cdot)$. More generally, UOT-RFM with correction order $k$ generates a distribution $p_1 \propto s_\tau^{-k} \pi_{\tau,1}^u = s_\tau^{-(k-1)} \nu$.*

*Proof.* As an overview, the proof relies on two observations: (1) training with $\pi^u$ yields $p_1 = \pi_1^u$, i.e., the biased UOT marginal (Theorem B.1) and (2) importance reweighting with $s_\tau^{-1}$ corrects this bias, since $\nu = s_\tau^{-1} \pi_1^u$ by the Radon–Nikodym derivative. Substituting this correction into the conditional flow matching loss yields Eq. 19, and hence the generated distribution recovers $\nu$.

Formally, Lemma C.3 shows that training a flow matching model with the optimal source-fixed UOT coupling $\pi_\tau^u$, i.e.,

$$\mathcal{L}_{\mathrm{UOT-CFM}}(\boldsymbol{\theta}) = \mathbb{E}_{t \sim \mathcal{U}[0,1], \mathbf{z} \sim \pi_\tau^u, \mathbf{x}_t \sim p_t(\mathbf{x}_t|\mathbf{z})} \|\mathbf{v}_\theta(t, \mathbf{x}_t) - \mathbf{u}_{t|\mathbf{z}}(\mathbf{x}_t|\mathbf{z})\|_2^2. \tag{20}$$

yields a flow matching model whose boundary conditions converge to $p_0 \to \pi_{\tau,0}^u, p_1 \to \pi_{\tau,1}^u$ as $\sigma \to 0$. By Theorem B.1, we have $\pi_{\tau,0}^u = \mu$ and $\pi_{\tau,1}^u \neq \nu$. Therefore, the UOT-CFM model generates a biased distribution.

Moreover, we now show that our UOT-RFM model with the first-order correction recovers the true target distribution. From Theorem B.1, we have $\pi_\tau^u \ll \nu$, so the Radon–Nikodym derivative exists and corresponds to the majority score. By our assumption $\nu \ll \pi_\tau^u$, it follows $\nu = s_\tau^{-1} \pi_1^u$. Therefore,

$$\mathcal{L}_{\text{ours},k=1}(\boldsymbol{\theta}) = \mathbb{E}_{t \sim \mathcal{U}[0,1], \mathbf{z} \sim \pi_\tau^u(\mathbf{z}), \mathbf{x}_t \sim p_t(\mathbf{x}_t|\mathbf{z})} \left[ s_\tau(\mathbf{x}_1)^{-1} \|\mathbf{v}_\theta(t, \mathbf{x}_t) - \mathbf{u}_{t|\mathbf{z}}(\mathbf{x}_t|\mathbf{z})\|_2^2 \right] \quad (21)$$

$$= \mathbb{E}_{t \sim \mathcal{U}[0,1], \mathbf{z} \sim \pi_\tau^u} \left[ s_\tau(\mathbf{x}_1)^{-1} \mathbb{E}_{\mathbf{x}_t \sim p_t(\mathbf{x}_t|\mathbf{z})} \left[ \|\mathbf{v}_\theta(t, \mathbf{x}_t) - \mathbf{u}_{t|\mathbf{z}}(\mathbf{x}_t|\mathbf{z})\|_2^2 \right] \right] \quad (22)$$

Note that the reweighted coupling $s_\tau(\mathbf{x}_1)^{-1} \pi_\tau^u(\mathbf{x}_0, \mathbf{x}_1)$ has the true target distribution $\nu$ as its marginal.

$$\int s_\tau(\mathbf{x}_1)^{-1} \pi_\tau^u(\mathbf{x}_0, \mathbf{x}_1) \, d\mathbf{x}_0 = s_\tau(\mathbf{x}_1)^{-1} \pi_{\tau,1}^u(\mathbf{x}_1) = \nu(\mathbf{x}_1). \quad (23)$$

Then, following a similar argument as in the UOT-CFM case, our UOT-RFM model with the first-order correction ($k = 1$) recovers the true target distribution $\nu$. Note that we specifically employ the source-fixed UOT coupling to ensure that the source marginal $\pi_{\tau,0}^u = \mu$ matches exactly with the initial distribution of the flow matching model. More generally, by a similar argument, UOT-RFM with correction order $k$ generates a distribution $p_1 \propto s_\tau^{-k} \pi_{\tau,1}^u$, up to a normalizing constant. $\qquad\square$

## D  RELATED WORKS

**Generative Models for Long-tailed Data**   Many real-world datasets follow long-tailed distributions, where a few dominant classes (Head class) contain the majority of samples, while numerous minority classes (Tail class) consist of a smaller number of samples. Generative modeling for long-tailed distributions often fails to learn the tail classes, resulting in low-diversity, low-quality samples. To address this, several GAN-based approaches have been proposed. CB-GAN (Rangwani et al., 2021) introduces a regularizer by utilizing a pretrained classifier to balance class learning. gSR-GAN (Rangwani et al., 2022) mitigates tail-class mode collapse through group spectral regularization. UTLO (Khorram et al., 2024) encourages head-to-tail knowledge sharing by combining an unconditional low-resolution generator with a conditional high-resolution generator. More recently, diffusion-based methods have been developed for long-tailed generation. CBDM (Qin et al., 2023) and LTDM (Zhang et al., 2024) improve tail-class quality by transferring knowledge from head to tail classes. In parallel, test-time guidance methods have been introduced. Um et al. (2024) uses proxy class labels to guide minority sampling. Um & Ye (2024) leverages a self-consistent minority score for minority guidance. In contrast, our method is label-free and corrects majority bias directly during training by leveraging the geometric structure of data via Unbalanced Optimal Transport. Furthermore, to the best of our knowledge, our approach is the first flow matching model designed for long-tailed distributions.

**Coupling in Flow Matching**   A key design choice in flow matching is the coupling between source and target samples. The original framework (Lipman et al., 2022) employs independent coupling. This independent coupling often produces curved trajectories due to flow crossing, increasing numerical errors and sampling cost (Park et al., 2024; Lee et al., 2023). To mitigate this, recent works proposed OT-based couplings between mini-batches (Pooladian et al., 2023; Tong et al., 2024) or trajectory refinement via pretrained models in Rectified Flow (Liu et al., 2023). Eyring et al. (2024) introduced the UOT-based couplings to the image-to-image translation, motivated by its adaptability to class imbalance. Model-Aligned Coupling (Lin et al., 2025) dynamically adjusts couplings during training, aligning them with the flow matching model being trained. However, these approaches do not tackle the majority bias in long-tailed generation. Our work complements them by combining UOT-based couplings with a reweighting mechanism that explicitly mitigates majority oversampling. This is the first attempt to utilize the density ratio between the target distribution and the UOT coupling as the majority score. We employ this score to weight the flow matching objective for long-tailed generation and formally characterize the resulting generated distribution (Theorem 3.1).

# E    IMPLEMENTATION DETAILS

This section provides the specific implementation details for our experiments on the CIFAR-10 and 2D synthetic datasets.

## E.1    EXPERIMENTS ON CIFAR-10

**Datasets**    We use two datasets for our image generation experiments: the standard CIFAR-10/100 dataset and their long-tailed version, CIFAR-10-LT and CIFAR-100-LT. The CIFAR-10-LT/100-LT are generated to simulate class imbalance, following an exponential decay distribution. The number of samples $n_i$ for each class $c_i$ is determined by the formula $n_i = \lfloor n_{\max} \cdot \mathcal{I}^{\frac{i}{M-1}} \rfloor$, where $M \in \{10, 100\}$ is the total number of classes, $n_{\max}$ is the number of samples in the largest class, and the imbalance factor $\mathcal{I}$ is set to 0.01.

**Network Architecture**    We employ the U-Net architecture provided in the `torchcfm` in Tong et al. (2024), without any modifications. The architecture uses four resolution levels with two residual blocks per level in both encoder and decoder, linked by skip connections at matching scales. Each block uses 3×3 convolutions with Group Normalization, SiLU activations, and dropout. Down-sampling is performed by stride-2 convolutions, and up-sampling uses nearest-neighbor interpolation followed by a 3×3 convolution.

**Training Details**    All experiments on CIFAR-10/100 follow the default settings of `torchcfm`. We use the `dopri5` ODE solver. For optimization, we use the Adam optimizer with a learning rate of $2 \times 10^{-4}$. The model is trained for a total of 400,000 iterations with a batch size of 128. Data preprocessing includes `transforms.RandomHorizontalFlip()` and normalization of pixel values to the range $[-1, 1]$ using `transforms.Normalize(mean=[0.5, 0.5, 0.5]`, `std=[0.5, 0.5, 0.5])`. For stable training, we apply a warmup schedule for the first 5,000 iterations, linearly increasing the learning rate from 0 to its target value, and use gradient clipping with an L2-norm threshold of 1.0. For Unbalanced Optimal Transport (UOT), the entropy regularization parameter $\epsilon$ is set to $5 \times 10^{-2}$, and the source marginal relaxation weight $\tau_1$ is set to infinity.

**Method Details**    The training process of our proposed method is as follows: (1) Sample mini-batches from each distribution. (2) Compute the coupling (transport plan) between the two mini-batches. (3) Determine the weight for each sample based on the computed transport plan. (4) Estimate the vector fields by feeding the coupled sample pairs into the U-Net and compute the weighted loss. (5) Update the network parameters via backpropagation. The specifics of each coupling method are as follows:

- **ICFM:** Uses an independent coupling, assuming the two distributions are independent.

- **OT-CFM:** Computes the transport plan $\pi$ using the `pot.emd` function and samples pairs according to the normalized probability distribution.

- **UOT-CFM:**    Computes    the    transport    plan    $\pi^u$    using    the `pot.unbalanced.sinkhorn_knopp_unbalanced` function and samples pairs based on the normalized probabilities.

- **UOT-RFM:** Also uses `pot.unbalanced.sinkhorn_knopp_unbalanced`, but samples only one target for each source sample from the normalized transport plan $\pi_\tau^u$.

The sample weights are calculated using the column sums of the transport plan $\pi$, which corresponds to the empirical measure of the target distribution, denoted as $\tilde{\nu}$. The weight $s_\tau^{-k}(\mathbf{x}_1)$ is defined as Eq. 8. The final loss function is the weighted mean squared error (MSE) between the vector fields: $\mathbb{E}_{(\mathbf{x}_0, \mathbf{x}_1) \sim \pi_\tau^u} \left[ s_\tau^{-k}(\mathbf{x}_1) \| \mathbf{v}_t(\mathbf{x}_0, \mathbf{x}_1) - \mathbf{u}_t(\mathbf{x}_0, \mathbf{x}_1) \|^2 \right]$.

**Evaluation Metrics**    To assess the quality of the generated images, we use the Fréchet Inception Distance (FID), Precision, and Recall. FID scores are calculated using the `cleanfid` library. For evaluation against the standard CIFAR-10/100 dataset, we use the library's built-in feature statistics. For CIFAR-10-LT and CIFAR-100-LT, the real data statistics are computed from a long-tailed dataset generated in the same manner as the training set. Precision and Recall are measured based on a

widely-used implementation[2], where the real data distribution is also generated identically to the training setup.

# F ADDITIONAL EXPERIMENTAL RESULTS ON MAJORITY BIAS

## F.1 CLASS DISTRIBUTION

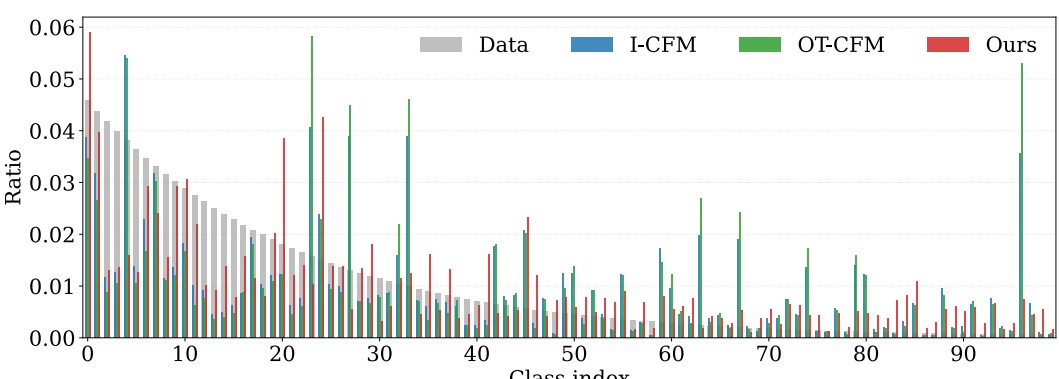

Figure 7: **Generated class distribution on CIFAR-100-LT** with $\mathcal{I} = 0.01$.

Figure 7 visualizes the distribution of the CIFAR-100-LT dataset and the classification results of samples generated by various generative models trained on the CIFAR-100-LT dataset. The classification was performed using a pre-trained classification model (RepVGG-A2) trained on CIFAR100. Ideally, a generative model should produce samples that follow the data's class distribution; however, the results show a notable divergence. Our model, UOT-RFM ($\tau = 2.0, k = 16.0$), can be seen in Figure 7 to better follow the class distribution of the data compared to the baselines I-CFM and OT-CFM.

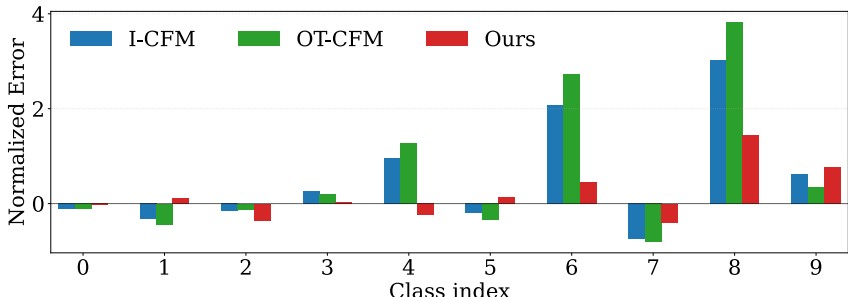

Figure 8: **Signed Normalized Class Ratio Error (Signed NCRE) between generated sample and CIFAR-10-LT. The average NCREs are 0.84 for I-CFM, 1.02 for OT-CFM, and 0.40 for our method.**

In Figure 8, we present a visualization of the Signed Normalized Class Ratio Error (Signed NCRE), calculated as $\frac{r_{gen,i} - r_{data,i}}{r_{data,i}}$ for each class, where a lower absolute value implies better alignment with the data distribution. Our analysis reveals that both I-CFM and OT-CFM exhibit significant discrepancies, while our proposed method shows notably better class distribution alignment (with $\tau = 1.0, k = 10.0$). This is quantitatively supported by the mean of NCRE, which are 0.84, 1.02, and 0.40 for I-CFM, OT-CFM, and our method, respectively.

---

[2]https://github.com/blandocs/improved-precision-and-recall-metric-pytorch

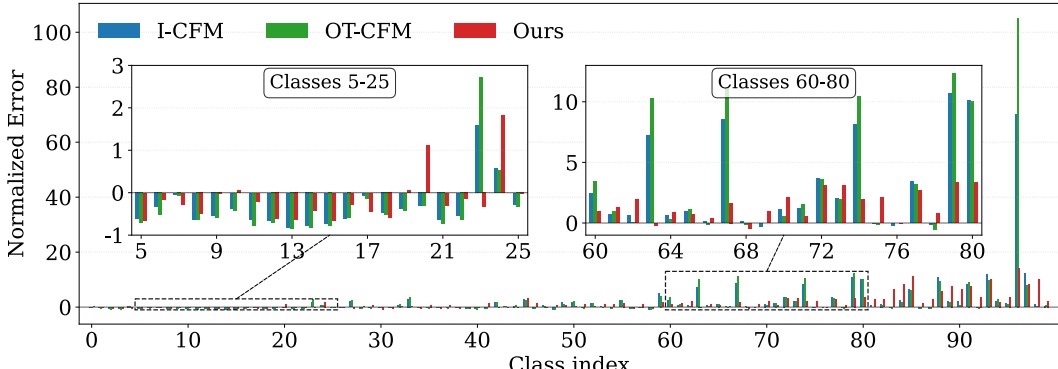

Figure 9: **Signed Normalized Class Ratio Error (Signed NCRE) between generated sample and CIFAR-100-LT**. The average NCREs are 2.41 for I-CFM, 2.79 for OT-CFM, and 1.82 for our method.

We visualized the Signed NCRE of CIFAR-100-LT for each class in Figure 9. While I-CFM and OT-CFM often show large discrepancies, our method ($\tau = 2.0, k = 16.0$) shows a much similar class distribution to that of data. The mean NCRE scores are 2.41 for I-CFM, 2.79 for OT-CFM, and 1.82 for our method, indicating superior distributional fidelity. The two zoomed-in subplots further illustrate the advantage of our approach, particularly in modeling minority classes. In the right subplot (classes 60–80), our method shows substantially lower normalized errors, while I-CFM and OT-CFM display large discrepancies for these rare classes. These visual and quantitative results together confirm that our method more accurately preserves the original class distribution, especially in the challenging tail regions.

### F.2 CLASSWISE NEGATIVE LOG-LIKELIHOOD (NLL)

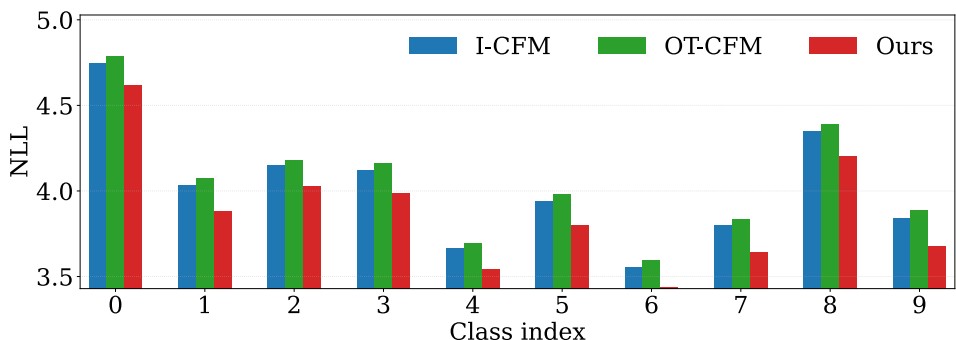

Figure 10: **Class-wise negative log-likelihood of CIFAR-10-LT** data samples on each model. Our method achieves an overall average NLL of 3.88, while I-CFM and OT-CFM achieve 4.02 and 4.06, respectively.

To measure how faithfully each model adheres to the distribution, we also measure the Negative Log-Likelihood (NLL). The log-likelihood of data sample $\mathbf{x}_1$ is computed by solving an ODE at each data sample as shown in the following equation:

$$\log p_1(\mathbf{x}_1) = \log p_0(\mathbf{x}_0) - \int_1^0 \mathrm{div}_t(\mathbf{v})dt,$$

where integral of divergence is approximated by accumulation of ODE simulation. In our experiments, we use the BPD(bits per dimension) for comparison as NLL $= -\log p_1(\mathbf{x}_1)/(\log(2) \cdot d)$, where $d$ denotes dimension of data.

To further analyze our model's performance on long-tailed distributions, we visualize the class-wise negative log-likelihood (NLL) on the CIFAR-10-LT (imbalance factor $\mathcal{I} = 0.01$) dataset in Figure 10.

Our method achieves the lowest (best) NLL across all ten classes, from index 0 to 9, which indicates that our model can more accurately estimate the distribution within each class. Our method achieves an overall mean NLL of 3.88, while I-CFM and OT-CFM achieve 4.02 and 4.06, respectively.

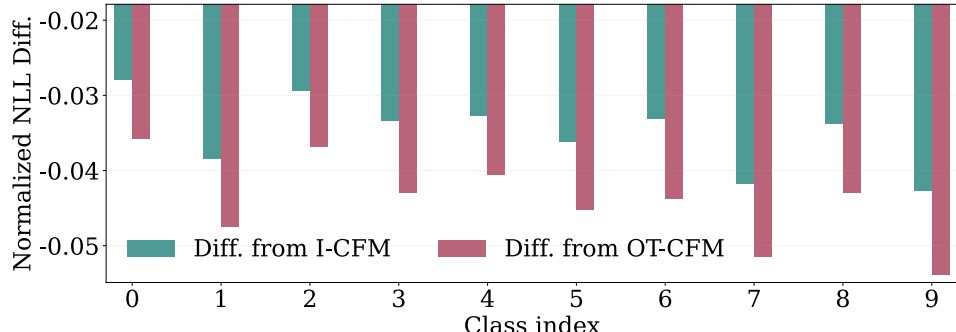

Figure 11: **Normalized difference of class-wise NLL between our model and the baselines on CIFAR-10-LT**.

To provide a more detailed view of this improvement, Figure 11 shows the normalized difference in NLL between our method and the baselines, I-CFM and OT-CFM. The results demonstrate that our NLL values are consistently lower than the baselines, with improvements ranging from a minimum of approximately 2.8% (class 0 and 2, from I-CFM) to a maximum of over 5.0% (class 7 and 9, from OT-CFM). Additionally, the difference is more pronounced in the tail classes. This provides clear evidence that our approach is more effective at capturing the true data distribution, particularly for the underrepresented classes in a long-tailed setting.

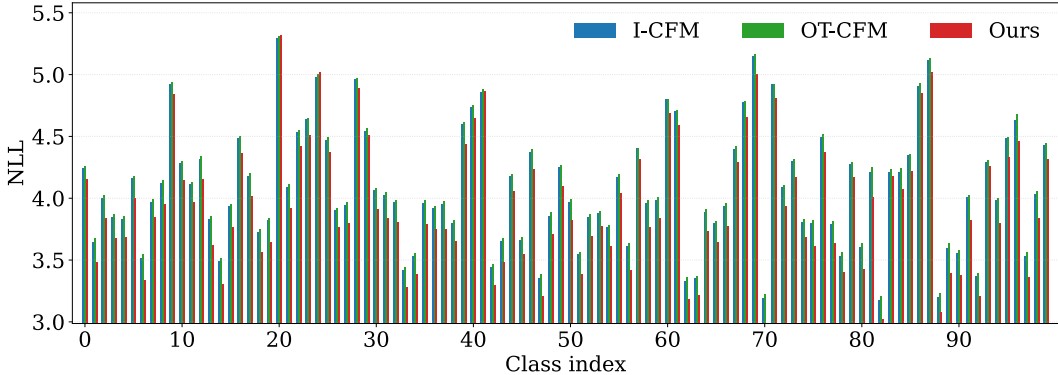

Figure 12: **Class-wise negative log-likelihood of CIFAR-100-LT** data samples on each model. Our method achieves an overall average NLL of 3.94, while I-CFM and OT-CFM achieve 4.08 and 4.10, respectively.

Similarly, we also measured and visualized the class-wise negative log-likelihood for CIFAR-100-LT (imbalance factor $\mathcal{I} = 0.01$). Figure 12 visualizes the average NLL(lower is better) for each class and the mean NLL. Our model achieves 3.94 mean NLL which is lower than I-CFM(4.08) and OT-CFM(4.10) in the CIFAR-100-LT dataset.

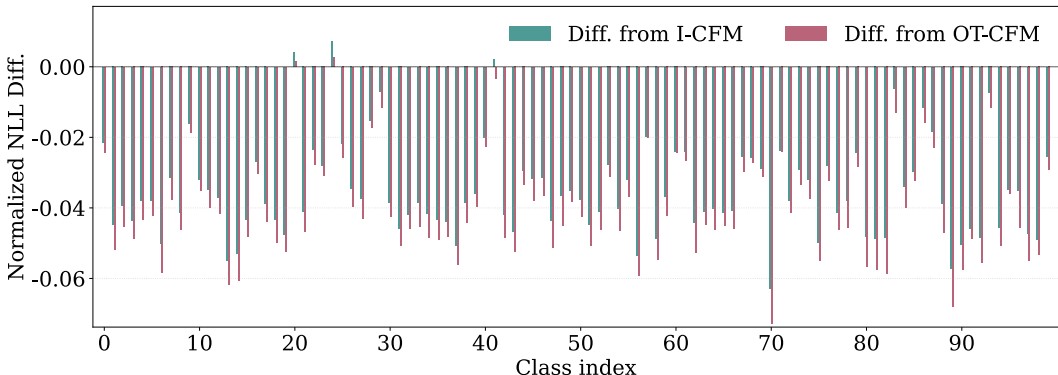

Figure 13: **Normalized difference of class-wise NLL between our model and the baselines on CIFAR-100-LT**.

We also show the normalized difference between our model and the competing models in Figure 13 to emphasize the gap. Our results show that our method achieves a lower(better) NLL than I-CFM and OT-CFM for most classes.

### F.3 CORRELATION BETWEEN MAJORITY SCORE AND HEAD-TAIL RELATIONSHIP

To evaluate whether the majority-score estimator reflects head–tail relationship, we conducted a synthetic data experiment. Specifically, we constructed a 3-component 2D Gaussian mixture as the target distribution, with clearly imbalanced mixture weights (Head: 0.70, medium-tail: 0.275, extreme tail: 0.025). The component mean vectors are set to $(0, 0)$ for head, $(4, 3)$ for medium-tail, and $(4, -3)$ for extreme tail. All components have the same covariance matrix $0.5^2 I$. For the source distribution, we used a single-mode Gaussian with zero mean $(0, 0)$ and covariance $0.5^2 I$. The majority score is computed using the unbalanced Sinkhorn solver (Flamary et al., 2021). Note that this is identical to how UOT-RFM estimates the majority score between mini-batches. (Flamary et al., 2021)

The results (Fig. 14 and Table 5) show a clear correspondence between the estimated majority score and the true head–tail dominance. Both the qualitative visualization and the average scores exhibit the consistent ordering:

$$\text{head} \ > \ \text{medium-tail} \ > \ \text{extreme-tail},$$

precisely matching the underlying mixture weights. This provides direct evidence that the majority-score estimator correctly captures head–tail structure.

Additionally, the effect of the marginal matching intensity $\tau$ is clearly reflected in the results. As $\tau$ increases, the optimal marginal $\pi_1$ in the UOT objective (Eq. 7) is forced to more closely match the data distribution $\nu$, causing the majority scores $s_\tau$ to cluster closer to 1 and reducing the head–tail separation. This trend is confirmed in Table 5.

Table 5: **Average majority score for each class** (Head, Medium-tail, and Extreme-tail). The majority score reflects the head-tail relationship. Also, as the marginal matching intensity $\tau$ increases, the spread between head and tail decreases.

| $\tau$ | Head | Medium-tail | Extreme-tail |
|---|---|---|---|
| $\tau = 1$ | 1.43 | 0.05 | 0.04 |
| $\tau = 3$ | 1.31 | 0.33 | 0.33 |
| $\tau = 5$ | 1.21 | 0.53 | 0.50 |
| $\tau = 10$ | 1.12 | 0.73 | 0.72 |

## G COMPARISON OF ODE SOLVERS

We compared the adaptive ODE solver (*dopri5*) with the Euler solver across a wide range of NFEs (number of function evaluations) (Table 6). Note that the dopri5 solver automatically selects an

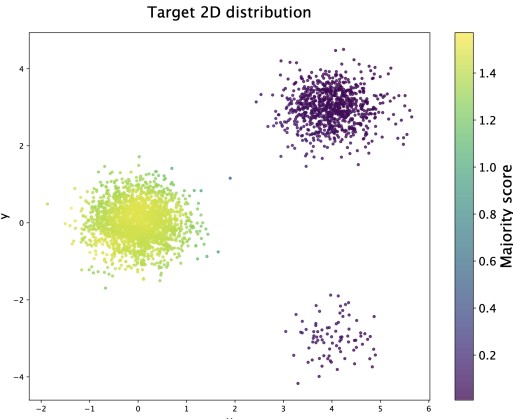

Figure 14: **Visualization of the majority score** $s_\tau(\cdot)$ on 3-component 2D Gaussian mixture target distribution, with imbalanced mixture weights (Head: 0.70, medium-tail: 0.275, extreme tail: 0.025) under $\tau = 1$. The source distribution is a single-mode Gaussian with mean $(0, 0)$ and covariance $0.5^2 I$. The majority score shows the average value of 1.43 for Head, 0.05 for Medium-tail, and 0.04 for Extreme-tail, reflecting the head–tail structure.

appropriate number of steps during inference. For example, for randomly selected samples, UOT-RFM and OT-CFM samples are generated under the same hyper-parameters atol($1e-5$) and rtol($1e-5$), which yield the comparable NFE of 152 and 140 respectively. With the Euler solver, the FID scores of both models gradually improve as the NFE increases, but UOT-RFM consistently outperforms OT-CFM across all NFEs.

Table 6: FID comparison between UOT-RFM and OT-CFM across different ODE solvers and NFEs.

| Method | UOT-RFM ($\downarrow$) | OT-CFM ($\downarrow$) |
|---|---|---|
| dopri5 (NFE: 152/140) | **11.16** | 17.58 |
| Euler (5 steps) | 38.00 | 39.13 |
| Euler (10 steps) | 19.61 | 24.99 |
| Euler (20 steps) | 14.74 | 20.07 |
| Euler (50 steps) | 12.37 | 17.84 |
| Euler (100 steps) | 11.72 | 17.37 |
| Euler (200 steps) | 11.49 | **17.18** |
| Euler (500 steps) | 11.28 | 17.27 |

## H  ADDITIONAL ABLATION STUDIES

Table 7: **Ablation study on the correction order $k$ and the marginal matching strength $\tau$.** Reported values are FID scores.

| $\tau \backslash k$ | LT→LT | | | | LT→Balanced | | | |
|---|---|---|---|---|---|---|---|---|
| | 1.0 | 2.0 | 4.0 | 8.0 | 1.0 | 2.0 | 4.0 | 8.0 |
| 2.0 | 13.77 | 13.41 | 12.42 | 11.37 | 25.02 | 24.60 | 24.54 | 24.76 |
| 4.0 | 14.01 | 13.78 | 13.68 | 12.67 | 24.94 | 24.65 | 24.45 | 24.37 |
| 6.0 | 14.39 | 13.72 | 13.48 | 12.41 | 24.91 | 24.88 | 24.86 | 24.90 |

Table 7 presents an ablation study on the effects of the correction order $k$ and the marginal matching strength $\tau$. All models were trained on the CIFAR-10-LT dataset. The "LT→LT" columns show FID scores measured against the CIFAR-10-LT dataset itself, assessing fidelity to the training distribution.

The "LT→Balanced" columns show FID scores using the class-balanced CIFAR10 dataset as a reference, evaluating the generation of a balanced distribution. First, analyzing the LT→LT results, the task is to faithfully replicate the long-tailed training distribution. In this scenario, a clear trend emerges: performance consistently improves as the correction order $k$ increases. For any given value of $\tau$, a larger $k$ leads to a lower (better) FID score. For example, when $\tau = 2.0$, the FID score monotonically decreases from 13.77 at $k = 1.0$ to a superior 11.37 at $k = 8.0$. This indicates that overcorrection ($k > 1$) is consistently beneficial, helping the model to more accurately estimate and represent the target long-tailed marginal distribution. In contrast, the LT→Balanced setting reveals a more complex trade-off. Here, a smaller $\tau$ (e.g., 2.0) enables a strong corrective weight but diminishes the sampling probability of minor classes. Conversely, a larger $\tau$ (e.g., 6.0) improves the sampling of these classes but flattens the weights, reducing their corrective impact. This necessitates a higher correction order $k$ to induce overcorrection. For instance, with $\tau = 4.0$, increasing $k$ from 1.0 to 8.0 improves the FID score from 24.94 to 24.37. However, excessive overcorrection can overshoot the balanced target, as seen for $\tau = 2.0$, where the FID score worsens from 24.54 ($k = 4.0$) to 24.76 ($k = 8.0$).

Table 8: **FID (↓) scores for CIFAR-10 based datasets**. The **boldface** and underlined values indicate the best and second-best performance. * indicates the best performance of UOT-RFM.

| model($k$) \ $\tau$ | CIFAR-10 | | | CIFAR-10-LT (0.01) | | | CIFAR-10-LT (0.001) | | |
|---|---|---|---|---|---|---|---|---|---|
| | 2.0 | 4.0 | 6.0 | 2.0 | 4.0 | 6.0 | 2.0 | 4.0 | 6.0 |
| I-CFM | —— | 3.78 | —— | —— | 14.39 | —— | —— | 17.54 | —— |
| OT-CFM | —— | 3.64 | —— | —— | 17.49 | —— | —— | 21.26 | —— |
| UOT-CFM | 3.62 | 3.72 | 3.79 | 14.31 | 14.37 | 14.41 | 19.16 | 18.13 | 18.23 |
| ours(1.0) | 3.62 | 3.71 | **3.58\*** | 13.97 | 14.01 | 14.37 | 17.15 | 18.16 | 17.89 |
| ours(2.0) | 3.80 | 3.60 | 3.70 | 13.22 | 13.76 | 13.55 | 16.96 | 17.12 | 17.49 |
| ours(4.0) | 4.12 | 3.87 | 3.82 | 12.36 | 13.73 | 13.62 | 16.02 | 16.52 | 17.34 |
| ours(6.0) | 4.51 | 3.92 | 3.82 | 11.60 | 13.58 | 13.23 | 15.13 | 16.80 | 16.89 |
| ours(8.0) | 4.64 | 4.63 | 3.92 | 11.41 | 12.81 | 12.34 | 13.32 | 16.24 | 15.87 |
| ours(10.0) | 5.21 | 4.36 | 3.92 | **11.07\*** | 11.96 | 12.90 | **12.84\*** | 15.05 | 16.06 |
| ours(16.0) | 7.28 | 4.63 | 4.20 | 12.76 | 11.37 | 12.32 | 13.25 | 13.36 | 14.64 |

Table 8 shows the Fréchet Inception Distance (FID) scores on three CIFAR-10 based datasets: CIFAR-10, CIFAR-10-LT ($\mathcal{I} = 0.01$), and CIFAR-10-LT ($\mathcal{I} = 0.001$). Our model consistently achieved the best FID score on each dataset. For the standard CIFAR-10 dataset, our model with $k = 1.0$ and $\tau = 6.0$ achieved the lowest FID score of 3.58, which is an improvement over the previous state-of-the-art model UOT-CFM's best score of 3.62. This improvement can be attributed to the correction of the intra-class feature majority bias. For the CIFAR-10-LT (0.01) and CIFAR-10-LT (0.001) datasets, which are long-tailed distributions, our model achieved the best scores of 11.07 and 12.84, respectively, with $k = 10.0$ and $\tau = 2.0$. These results represent a significant improvement over the baseline models I-CFM and UOT-CFM, which achieved 14.39 and 17.54 respectively.

Table 9: **Precision (↑) scores for CIFAR-10 based datasets.**

| model($k$) \ $\tau$ | CIFAR-10 | | | CIFAR-10-LT (0.01) | | |
|---|---|---|---|---|---|---|
| | 2.0 | 4.0 | 6.0 | 2.0 | 4.0 | 6.0 |
| I-CFM | —— | 0.46 | —— | —— | 0.68 | —— |
| OT-CFM | —— | 0.47 | —— | —— | **0.71** | —— |
| UOT-CFM | 0.46 | 0.46 | 0.47 | 0.67 | 0.67 | 0.67 |
| ours(1.0) | 0.46 | 0.46 | 0.47 | 0.65 | 0.65 | 0.65 |
| ours(2.0) | 0.47 | 0.46 | 0.47 | 0.64 | 0.65 | 0.66 |
| ours(4.0) | 0.47 | 0.46 | 0.46 | 0.63 | 0.65 | 0.65 |
| ours(6.0) | 0.47 | 0.46 | 0.46 | 0.65 | 0.65 | 0.65 |
| ours(8.0) | 0.47 | 0.47 | 0.46 | 0.63 | 0.64 | 0.64 |
| ours(10.0) | 0.47 | 0.47 | 0.46 | 0.61 | 0.62 | 0.62 |
| ours(16.0) | **0.48** | 0.47 | 0.46 | 0.63 | 0.62 | 0.63 |

In addition to the FID scores, we conducted a quantitative analysis using precision and recall metrics, with the results summarized in Tables 9 and 10. The precision scores generally remain comparable (CIFAR-10) to or lower (CIFAR-10-LT) than the baselines, while the recall scores show marked improvements (CIFAR-10-LT) in some cases. This behavior is particularly evident on the CIFAR-10-LT (0.01) dataset. For example, our model with $k = 10.0$ and $\tau = 4.0$ achieves a recall of 0.41, a significant improvement over the best baseline score of 0.29 from I-CFM and UOT-CFM. This enhancement in recall highlights our model's increased capability to generate diverse samples that cover the full spectrum of the data distribution, especially the minority classes.

The F1 score, which harmonizes precision and recall, serves as a comprehensive metric for evaluating generative models, and our model consistently shows better performance in cases where recall is improved, as shown in Table 11. Specifically, on the CIFAR-10-LT (0.01) dataset, our models such as a setting with $k = 10.0$ and $\tau = 4.0$ achieve an F1 score of 0.49, which is higher than the best

Table 10: **Recall (↑) scores for CIFAR-10 based datasets.**

| model($k$) \ $\tau$ | CIFAR-10 | | | CIFAR-10-LT (0.01) | | |
|---|---|---|---|---|---|---|
| | 2.0 | 4.0 | 6.0 | 2.0 | 4.0 | 6.0 |
| I-CFM | —— 0.38 —— | | | —— 0.29 —— | | |
| OT-CFM | —— 0.38 —— | | | —— 0.24 —— | | |
| UOT-CFM | 0.38 | 0.38 | 0.38 | 0.28 | 0.29 | 0.28 |
| ours(1.0) | 0.38 | 0.38 | 0.38 | 0.27 | 0.28 | 0.28 |
| ours(2.0) | 0.38 | 0.39 | 0.38 | 0.29 | 0.28 | 0.28 |
| ours(4.0) | 0.38 | 0.38 | **0.39** | 0.29 | 0.29 | 0.28 |
| ours(6.0) | 0.38 | 0.38 | **0.39** | 0.28 | 0.28 | 0.28 |
| ours(8.0) | 0.38 | 0.38 | 0.38 | 0.33 | 0.30 | 0.29 |
| ours(10.0) | 0.38 | 0.38 | 0.38 | 0.38 | **0.41** | **0.41** |
| ours(16.0) | 0.36 | 0.38 | **0.39** | 0.32 | 0.32 | 0.31 |

Table 11: **F1 (↑) scores for CIFAR-10 based datasets.**

| model($k$) \ $\tau$ | CIFAR-10 | | | CIFAR-10-LT (0.01) | | |
|---|---|---|---|---|---|---|
| | 2.0 | 4.0 | 6.0 | 2.0 | 4.0 | 6.0 |
| I-CFM | —— 0.42 —— | | | —— 0.41 —— | | |
| OT-CFM | —— 0.42 —— | | | —— 0.36 —— | | |
| UOT-CFM | 0.42 | 0.42 | 0.41 | 0.40 | 0.42 | 0.40 |
| ours(1.0) | 0.42 | 0.42 | 0.42 | 0.38 | 0.39 | 0.39 |
| ours(2.0) | 0.42 | 0.41 | 0.41 | 0.40 | 0.40 | 0.40 |
| ours(4.0) | 0.42 | 0.42 | **0.43** | 0.40 | 0.41 | 0.40 |
| ours(6.0) | 0.42 | 0.41 | 0.42 | 0.40 | 0.40 | 0.40 |
| ours(8.0) | 0.42 | 0.42 | 0.41 | 0.46 | 0.41 | 0.40 |
| ours(10.0) | 0.42 | 0.42 | 0.41 | 0.47 | **0.49** | **0.49** |
| ours(16.0) | 0.41 | 0.42 | 0.42 | 0.43 | 0.43 | 0.42 |

baseline score of 0.42 from UOT-CFM. This demonstrates that even if there is no gain in precision, the increase in recall from our method's ability to better capture minority features leads to a more balanced and representative learned distribution overall.

Table 12: **FID ($\downarrow$) scores for CIFAR-100 based datasets.**

| model($k$) $\setminus \tau$ | CIFAR-100 | | | CIFAR-100-LT (0.01) | | | CIFAR-100-LT (0.001) | | |
|---|---|---|---|---|---|---|---|---|---|
| | 2.0 | 4.0 | 6.0 | 2.0 | 4.0 | 6.0 | 2.0 | 4.0 | 6.0 |
| I-CFM | —— | 6.39 | —— | —— | 25.56 | —— | —— | 31.86 | —— |
| OT-CFM | —— | **6.14** | —— | —— | 31.90 | —— | —— | 38.37 | —— |
| UOT-CFM | 6.48 | 6.63 | 6.45 | 26.96 | 25.78 | 25.33 | 33.32 | 31.89 | 31.83 |
| ours(1.0) | 6.77 | 6.54* | 6.55 | 25.11 | 25.84 | 25.07 | 32.67 | 32.29 | 31.98 |
| ours(2.0) | 6.79 | 6.65 | 6.79 | 22.93 | 24.68 | 24.62 | 31.23 | 31.73 | 31.13 |
| ours(4.0) | 7.27 | 6.69 | 6.67 | 20.09 | 23.45 | 23.14 | 28.70 | 30.62 | 30.37 |
| ours(6.0) | 7.68 | 6.93 | 7.06 | 17.88 | 22.54 | 22.94 | 26.01 | 29.13 | 30.21 |
| ours(8.0) | 8.70 | 7.20 | 7.29 | 16.49 | 20.01 | 22.49 | 23.04 | 28.43 | 28.91 |
| ours(10.0) | 10.12 | 7.60 | 7.18 | **15.38*** | 18.74 | 21.33 | 20.04 | 25.35 | 28.18 |
| ours(16.0) | 15.90 | 8.73 | 7.57 | 17.22 | 16.00 | 17.90 | **18.40*** | 22.26 | 26.19 |

Table 12 presents the FID scores for CIFAR-100-based datasets. Our model demonstrates a significant performance boost on the long-tailed datasets compared to the baselines. On CIFAR-100-LT ($\mathcal{I} = 0.01$), our best score is 15.38 (with $k = 10.0, \tau = 2.0$), a substantial improvement over the best baseline of 25.33 from UOT-CFM. For the even more imbalanced CIFAR-100-LT ($\mathcal{I} = 0.001$) dataset, our model achieves a score of 18.40 (with $k = 16.0, \tau = 2.0$), which is a large improvement over the best baseline score of 31.83. This demonstrates our method's ability to effectively correct the majority bias present in these imbalanced datasets.

However, on the balanced CIFAR-100 dataset, our model's FID score is slightly higher than that of the competitive models. The best FID score for our method is 6.54, slightly underperforming the best baseline score of 6.14 from OT-CFM. This suggests that for a balanced dataset, a large correction order $k$ may not be optimal. Instead, very delicate hyperparameter tuning is likely required to achieve superior results. It is also worth noting that our approach, based on UOT-CFM, exhibits a relatively high FID score on CIFAR-100, where UOT-CFM itself has a score of 6.45. This can be attributed to the influence of UOT coupling. We leave the further refinement of the UOT coupling's effect on balanced datasets as a direction for future work.

Table 13: **Precision ($\uparrow$) scores for CIFAR-100 based datasets.**

| model($k$) $\setminus \tau$ | CIFAR-100 | | | CIFAR-100-LT (0.01) | | |
|---|---|---|---|---|---|---|
| | 2.0 | 4.0 | 6.0 | 2.0 | 4.0 | 6.0 |
| I-CFM | —— | 0.41 | —— | —— | 0.62 | —— |
| OT-CFM | —— | **0.43** | —— | —— | **0.73** | —— |
| UOT-CFM | 0.41 | 0.41 | 0.41 | 0.72 | **0.73** | 0.72 |
| ours(1.0) | 0.41 | 0.41 | 0.42 | 0.72 | **0.73** | 0.72 |
| ours(2.0) | 0.41 | 0.41 | 0.41 | 0.71 | 0.72 | 0.72 |
| ours(4.0) | 0.42 | 0.42 | 0.41 | 0.71 | 0.72 | 0.71 |
| ours(6.0) | 0.42 | 0.42 | 0.41 | 0.69 | 0.71 | 0.72 |
| ours(8.0) | 0.41 | 0.41 | 0.41 | 0.69 | 0.71 | 0.72 |
| ours(10.0) | 0.42 | 0.41 | 0.41 | 0.68 | 0.71 | 0.71 |
| ours(16.0) | **0.43** | 0.42 | 0.41 | 0.65 | 0.69 | 0.70 |

Tables 13, 14, and 15 present the precision, recall, and F1 scores for the CIFAR-100-based datasets. Our method shows a clear improvement in recall, especially on the CIFAR-100-LT ($\mathcal{I} = 0.01$) dataset. In the best case, with $k = 16.0$ and $\tau = 4.0$, our model achieves a recall score of 0.32, which is notably higher than the best baseline recall of 0.29 from I-CFM. This recall improvement indicates our model's enhanced ability to generate more diverse samples, particularly for the minority classes.

Table 14: **Recall (↑) scores for CIFAR-100 based datasets.**

| | CIFAR-100 | | | CIFAR-100-LT (0.01) | | |
|---|---|---|---|---|---|---|
| model($k$) $\backslash \tau$ | 2.0 | 4.0 | 6.0 | 2.0 | 4.0 | 6.0 |
| I-CFM | ——— **0.37** ——— | | | ——— 0.29 ——— | | |
| OT-CFM | ——— 0.36 ——— | | | ——— 0.24 ——— | | |
| UOT-CFM | **0.37** | 0.36 | 0.36 | 0.28 | 0.27 | 0.27 |
| ours(1.0) | 0.36 | 0.36 | 0.36 | 0.27 | 0.27 | 0.27 |
| ours(2.0) | 0.36 | 0.36 | 0.36 | 0.28 | 0.28 | 0.29 |
| ours(4.0) | 0.35 | 0.36 | 0.35 | 0.27 | 0.28 | 0.29 |
| ours(6.0) | 0.35 | 0.35 | 0.36 | 0.28 | 0.29 | 0.29 |
| ours(8.0) | 0.34 | 0.36 | 0.36 | 0.29 | 0.30 | 0.28 |
| ours(10.0) | 0.33 | 0.35 | 0.36 | 0.30 | 0.30 | 0.29 |
| ours(16.0) | 0.30 | 0.34 | 0.36 | 0.30 | **0.32** | 0.31 |

Table 15: **F1 (↑) scores for CIFAR-100 based datasets.**

| | CIFAR-100 | | | CIFAR-100-LT (0.01) | | |
|---|---|---|---|---|---|---|
| model($k$) $\backslash \tau$ | 2.0 | 4.0 | 6.0 | 2.0 | 4.0 | 6.0 |
| I-CFM | ——— **0.39** ——— | | | ——— 0.40 ——— | | |
| OT-CFM | ——— **0.39** ——— | | | ——— 0.36 ——— | | |
| UOT-CFM | **0.39** | 0.38 | 0.38 | 0.40 | 0.40 | 0.40 |
| ours(1.0) | 0.38 | 0.38 | **0.39** | 0.40 | 0.40 | 0.40 |
| ours(2.0) | 0.38 | 0.38 | 0.38 | 0.40 | 0.40 | 0.41 |
| ours(4.0) | 0.38 | **0.39** | 0.38 | 0.40 | 0.41 | 0.41 |
| ours(6.0) | 0.38 | 0.38 | **0.39** | 0.40 | 0.41 | 0.41 |
| ours(8.0) | 0.37 | 0.38 | 0.38 | 0.41 | 0.42 | 0.40 |
| ours(10.0) | 0.37 | 0.38 | 0.38 | 0.41 | 0.42 | 0.40 |
| ours(16.0) | 0.36 | 0.38 | 0.38 | 0.41 | **0.44** | 0.42 |

The F1 score, which balances precision and recall, also demonstrates our method's superiority in this setting. On the CIFAR-100-LT ($\mathcal{I} = 0.01$) dataset, our model with $k = 16.0$ and $\tau = 4.0$ achieves an F1 score of 0.44. This is a clear improvement over the best baseline F1 score of 0.40 from I-CFM and UOT-CFM, showing that our method's emphasis on minority features leads to a more balanced and accurate generative process.

However, our method shows slightly lower results on the CIFAR-100. In this case, our approach requires more delicate coupling and tuning rather than aggressive re-weighting to perform optimally, similar to the CIFAR-10 case.

# I    ADDITIONAL QUALITATIVE EXAMPLES

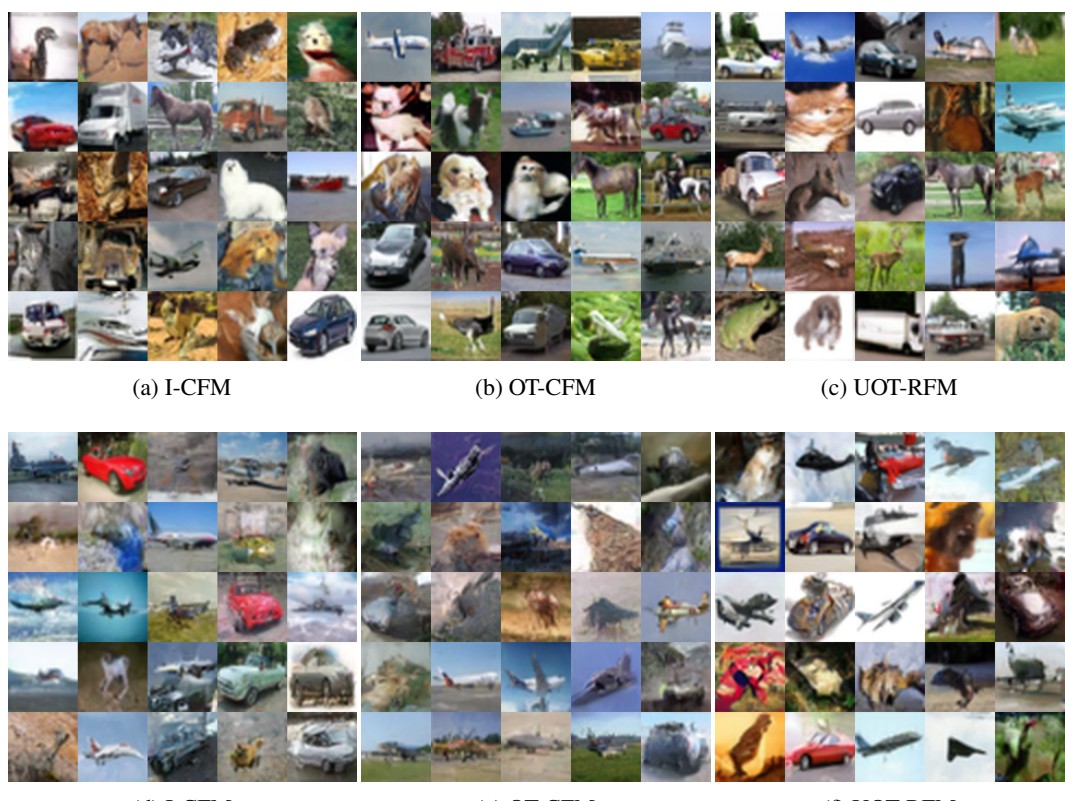

|        (a) I-CFM        |        (b) OT-CFM        |        (c) UOT-RFM        |

|        (d) I-CFM        |        (e) OT-CFM        |        (f) UOT-RFM        |

Figure 15: **CIFAR image generation results.** The first row shows images randomly generated from models trained on the balanced CIFAR10 dataset. The second row shows images from models trained on the CIFAR-10-LT dataset.

To qualitatively verify our models, we include images actually generated by each model in this paper. Figure 15 shows 25 completely randomly sampled generated images, without cherry-picking. The first row displays generated images from each model trained on the balanced CIFAR-10 dataset, and the second row shows generated images from each model trained on the long-tailed CIFAR-10-LT (0.01) dataset. Our method, UOT-RFM, utilizes hyperparameters $\tau = 1.0$ and $k = 1.0$ when trained on CIFAR-10, and $\tau = 1.0$ and $k = 10.0$ when trained on CIFAR-10-LT.

On the CIFAR-10 dataset, our model appears comparable to the baseline models I-CFM and OT-CFM. However, on the CIFAR-10-LT dataset, I-CFM and OT-CFM tend to generate images that are somewhat blurry and noisy. In contrast, our generated images are relatively clean and distinct.

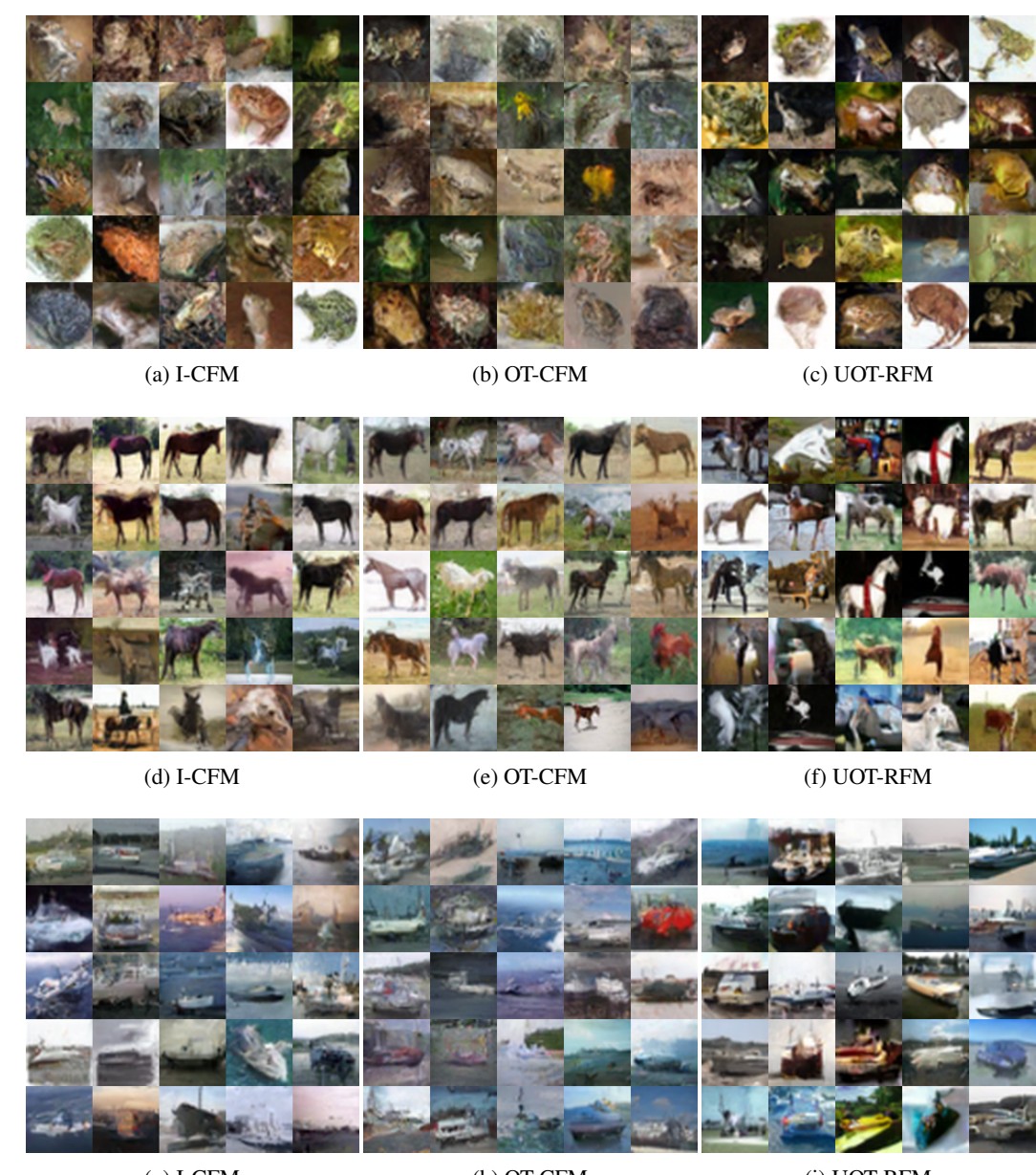

(a) I-CFM        (b) OT-CFM        (c) UOT-RFM

(d) I-CFM        (e) OT-CFM        (f) UOT-RFM

(g) I-CFM        (h) OT-CFM        (i) UOT-RFM

Figure 16: **Tail-classes CIFAR image generation results.** The classified images from models trained on the CIFAR-10-LT dataset.

Figure 16 visualizes the generated images for the tail classes of the CIFAR-10-LT dataset. Specifically, the first row shows images generated for class 06 (frog), the second row for class 07 (horse), and the last row for class 08 (ship). Note that the images for the last tail class, truck, were previously shown in the main text (Figure 5). The samples in each grid image are selected by the top-confident values from a classification model, which was pre-trained on the balanced CIFAR-10 dataset. Observing the results, the images generated by I-CFM and OT-CFM show a tendency to be relatively noisy. In contrast, our method, UOT-RFM, yields images that are cleaner and exhibit greater diversity within the class.

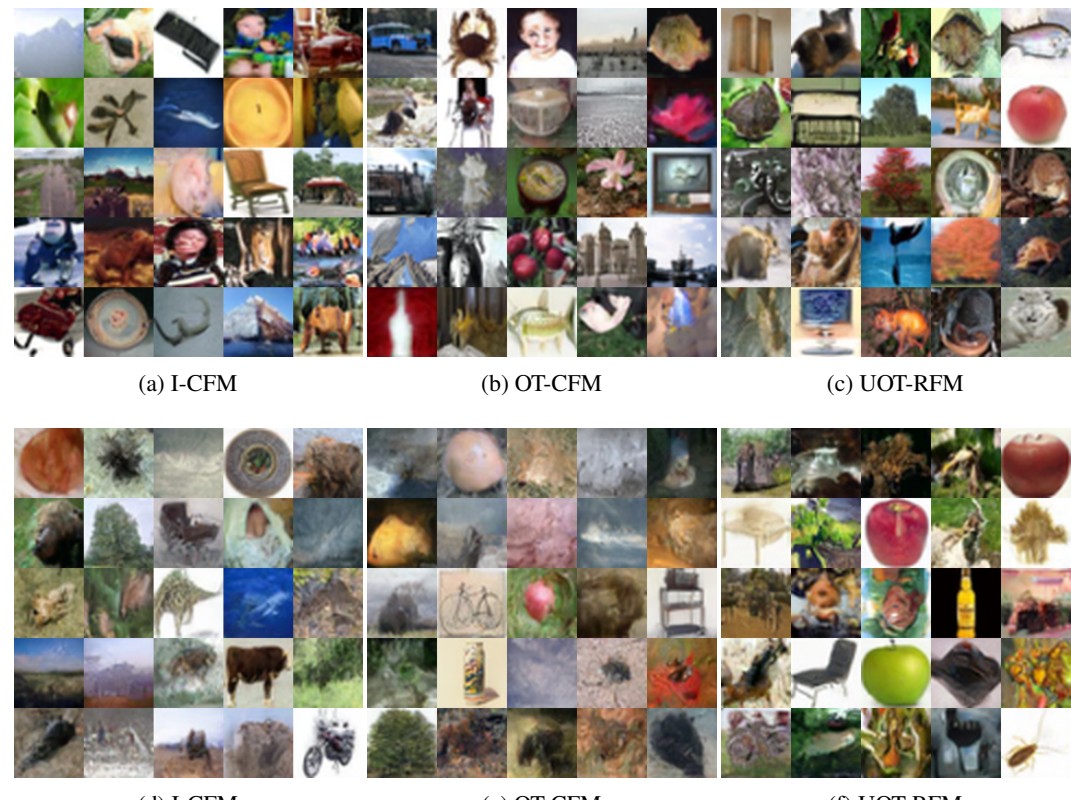

(a) I-CFM          (b) OT-CFM          (c) UOT-RFM

(d) I-CFM          (e) OT-CFM          (f) UOT-RFM

Figure 17: **CIFAR image generation results.** The first row shows images randomly generated from models trained on the balanced CIFAR100 dataset. The second row shows images from models trained on the CIFAR-100-LT dataset.

Figure 17 shows 25 completely randomly sampled images. The first row displays images generated when trained on the balanced CIFAR-100 dataset, and the second row shows images generated when trained on the long-tailed CIFAR-100-LT (0.01) dataset. The overall trend observed in CIFAR-100 is similar to that in CIFAR-10. Specifically, on the balanced CIFAR-100 dataset, our UOT-RFM method generates images that appear comparable to those from I-CFM and OT-CFM. However, when trained on the long-tailed CIFAR-100-LT, the generated images from I-CFM and OT-CFM tend to be blurry, whereas the images produced by UOT-RFM are noticeably cleaner and more distinct.

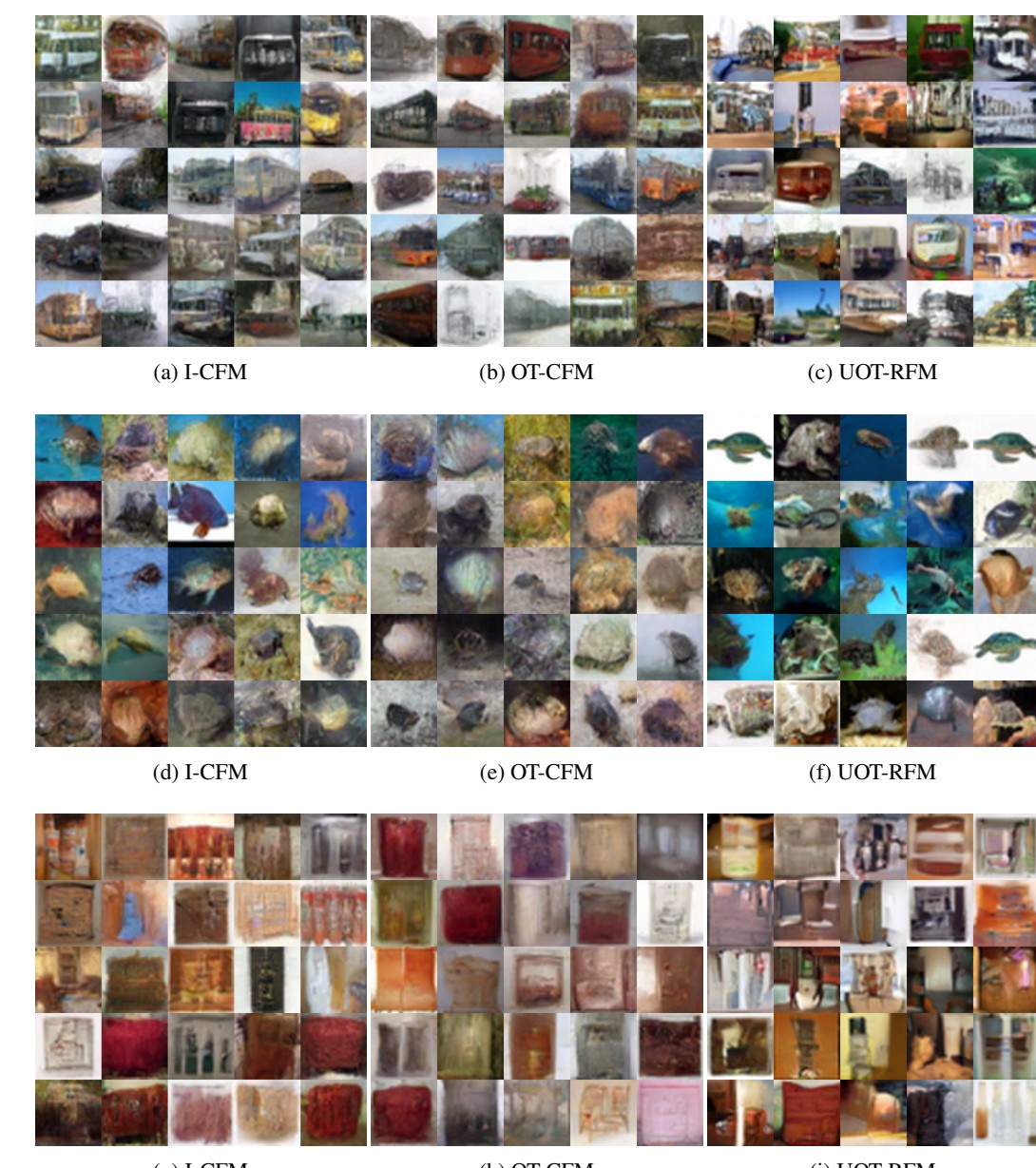

(a) I-CFM  (b) OT-CFM  (c) UOT-RFM

(d) I-CFM  (e) OT-CFM  (f) UOT-RFM

(g) I-CFM  (h) OT-CFM  (i) UOT-RFM

Figure 18: **Tail-classes CIFAR image generation results.** The classified images from models trained on the CIFAR-100-LT dataset.

Figure 18 visualizes the generated images for three specific tail classes from the CIFAR-100-LT dataset as examples. The first row shows images generated for class 81 (streetcar), the second row for class 93 (turtle), and the last row for class 94 (wardrobe). The samples in each grid image are selected by the top-confident values of a classification model. In this comparison, I-CFM and OT-CFM, particularly the latter, show a pronounced tendency to produce blurry and noisy images. In contrast, our UOT-RFM method produces images that are noticeably sharper and capture a better variety of features, such as shape, within each class.

