# OpenReview forum: "Reweighted Flow Matching via Unbalanced OT for Label-free Long-tailed Generation"
_ICLR.cc/2026/Conference — Submitted to ICLR 2026_

### Official Review · Reviewer_msMD · 2025-10-25

**Soundness:** 2
**Presentation:** 2
**Contribution:** 2
**Rating:** 4
**Confidence:** 4

**Summary:**

This paper proposed a re-weighting strategy via Unbalanced Optimal Transport (OT) for Label-free Long-tailed Generation. The main contribution of this paper is to leverage the unbalanced OT theory to quantify the degree of majority based on the geometric structure of the data, without requiring class labels. Then, this score can be used as the weight to balance the data distribution during training. The experimental results demonstrate the the proposed method can successfully alleivate the long-tailed problem.

**Strengths:**

1. The topic of this paper is important. The long-tailed problem exists during the training. Alleviating such a problem is essential.

2. This paper connects the OT and unbalanced OT to improve the flow-matching paradigm, which is interesting.

3. The experimental results demonstrate the validity of the proposed method.

**Weaknesses:**

The major weakness of this paper is that the experiments are not sufficient.

Concretely,

1) Lacking some related work that alleviates the long-tailed generation for diffusion [1]. Although the paradigm shifted from diffusion probabilistic models (DPMs) to FM, the ODE has changed, and FM`s FID is better than DPM. But the author can show the FID gain to further enhance their contribution.

2) Lacking the necessary experimental setting. For example, how many timesteps are used in the reverse process in this paper? The FID on CIFAR-10 is over 10, which is high. To better illustrate that the proposed method can alleviate the long-tailed problem, the author can make an ablation study about timesteps (E.g., T=50, T=100, T=200, T=500) to show the upper bound of the model.

3) Lacking the necessary ablation study for solvers. This paper used the dopri5 ODE solver, but the mainstream solver in FM is the Euler solver [2]. The author should present an ablation study to illustrate why they abandoned the mainstream solver.

[1] LONG-TAILED DIFFUSION MODELS WITH ORIENTED CALIBRATION. Zhang et al. ICLR 2024.

[2] Scaling Rectified Flow Transformers for High-Resolution Image Synthesis. Patrick Esser et al. ArXiv:2403.03200

**Questions:**

Then, I have two questions about this paper:

1. How to prove that the increase in FID is from the increase in diversity after alleviating the long-tailed problem instead of from the increase in image quality in the head classifier?  The motivation for this question comes from Fig. 4 and Fig. 7, which show that the proposed method appears to increase the model's likelihood of generating a head classifier sample. Since these samples dominate the dataset, the model will tend to perform better for them. This can also lead to an increase in FID since FID measures both the diversity and quality of images.

2. Does the proposed method support the CFG? Although this paper focuses on the unlabeled setting. CFG remains an important strategy that can be applied in an unlabeled setting. For example, the condition is an image too. Thus, we have to explore the influence of the proposed method on CFG.

To sum up, the proposed method in this paper is interesting, combining OT and UOT into the FM. But the experiments are not sufficient to support this paper. Meanwhile, there remain some questions that will influence the feasibility of the proposed method for training large diffusion models, such as i2i diffusion (e.g., SD 3.5). Therefore, I rate it as marginally below the acceptance threshold. But would not mind if paper is accepted. If the author could clarify these concerns, I am willing to increase my score.

---

> ### Author Response · Authors · 2025-11-23
> **Response to Reviewer msMD (1/2)**
>
> We sincerely thank the reviewer for carefully reading our manuscript and providing valuable feedback. We hope our responses to be helpful in addressing the reviewer's concerns. We highlighted the corresponding revisions in the manuscript in Blue.
>
> $ $
>
> ---
> > [W1] Lacking some related work that alleviates the long-tailed generation for diffusion [1]. Although the paradigm shifted from diffusion probabilistic models (DPMs) to FM, the ODE has changed, and FM`s FID is better than DPM. But the author can show the FID gain to further enhance their contribution.
>
> **A.** Thank you for suggesting the related work of LTDM [1]. As discussed in the introduction (Lines 48-50) and the related works (Lines 790-792), **LTDM relies on explicit class labels to improve tail-class generation. In contrast, our method adopts the label-free approach**. Because LTDM addresses a different problem setup with stronger assumptions, we view it as outside the scope of our approach. Nonetheless, we appreciate the suggestion and clarified this distinction more explicitly in the revision.
>
> $ $
>
> ---
> > [W2] Lacking the necessary experimental setting. For example, how many timesteps are used in the reverse process in this paper? The FID on CIFAR-10 is over 10, which is high. To better illustrate that the proposed method can alleviate the long-tailed problem, the author can make an ablation study about timesteps (E.g., T=50, T=100, T=200, T=500) to show the upper bound of the model.
>
> > [W3] Lacking the necessary ablation study for solvers. This paper used the dopri5 ODE solver, but the mainstream solver in FM is the Euler solver [2]. The author should present an ablation study to illustrate why they abandoned the mainstream solver.
>
> **A.**  We appreciate the reviewer for the thoughtful comment. First of all, we would like to clarify that UOT-RFM achieves **the FID score of 3.58 on CIFAR-10** (Table 4) and **11.03 on CIFAR-10-LT (imbalance ratio 0.01)** (Table 1). CIFAR-10-LT is a significantly more challenging setting than standard CIFAR-10, where the extreme-tail class has only 50 samples.
>
> Following the reviewer’s advice, we conducted additional experiments comparing the **adaptive ODE solver (dopri5)** with the **Euler solver** across a wide range of NFEs (number of time steps). The results are as follows:
>
> | Method | NFE | UOT-RFM FID | OT-CFM FID |
> | --- | --- | --- | --- |
> | dopri5 | 152 / 140 | 11.16 | 17.58 |
> | Euler | 5 | 38.00 | 39.13 |
> | Euler | 10 | 19.61 | 24.99 |
> | Euler | 20 | 14.74 | 20.07 |
> | Euler | 50 | 12.37 | 17.84 |
> | Euler | 100 | 11.72 | 17.37 |
> | Euler | 200 | 11.49 | 17.18 |
> | Euler | 500 | 11.28 | 17.27 |
>
> Note that the dopri5 solver automatically selects an appropriate number of steps during inference. **UOT-RFM and OT-CFM samples are generated through the comparable steps of 152 and 140 on average** with the dopri5 solver. With the Euler solver, the **FID scores of both models gradually improve as the NFE increases**, but UOT-RFM consistently outperforms OT-CFM across all NFEs. We included this comparison in the revised manuscript in Appendix G.

---

> > ### Author Response · Authors · 2025-11-23
> > **Response to Reviewer msMD (2/2)**
> >
> > ---
> >
> > > [Q1] How to prove that the increase in FID is from the increase in diversity after alleviating the long-tailed problem instead of from the increase in image quality in the head classifier? The motivation for this question comes from Fig. 4 and Fig. 7, which show that the proposed method appears to increase the model's likelihood of generating a head classifier sample. Since these samples dominate the dataset, the model will tend to perform better for them. This can also lead to an increase in FID since FID measures both the diversity and quality of images.
> >
> > **A.** We appreciate the reviewer for the insightful comment. The concern is whether the improved FID might come simply from generating more head-class samples rather than from genuinely improving diversity and tail-class fidelity. **Two class-wise evaluations (NCRE and class-wise NLL) directly address this issue**.
> >
> > First, **our model does not over-generate head-class samples but instead more accurately matches the true long-tailed distribution**. The Normalized Class Ratio Error (NCRE) is defined as follows:
> > $$
> >     \text{NCRE}\_i = \frac{|r\_{gen, i} - r\_{data, i}|}{r\_{data, i}},
> > $$
> > where $r_{gen, i}$ denotes the proportion of generated images assigned to class $c_i$ (via proxy labels) and $r_{data, i}$ is the ground-truth class proportion. Because the NCRE metric is normalized to the ground-truth class proportion, the proportion error in tail classes is penalized more severely. If our model were oversampling head classes, the NCRE for tail classes would increase sharply. However, as shown in Figs 8 and 9, our model achieves **lower NCRE across almost all classes**, especially for the tail classes on CIFAR-10-LT and CIFAR-100-LT. This shows that our method is not simply increasing head-class generation.
> >
> > Second, the **class-wise NLL metric directly evaluates the fidelity of generated samples within each class**. Figs 11 and 13 show that our model consistently achieves **better NLL scores across nearly all classes** on CIFAR-10-LT and CIFAR-100-LT. This result demonstrates that the improved FID is accompanied by higher per-class fidelity.
> >
> > Together, these results show that the FID improvement comes from better distributional alignment and improved tail-class fidelity, rather than from an overemphasis on head-class image quality.
> >
> > $ $
> >
> > ---
> >
> > > [Q2] Does the proposed method support the CFG? Although this paper focuses on the unlabeled setting. CFG remains an important strategy that can be applied in an unlabeled setting. For example, the condition is an image too. Thus, we have to explore the influence of the proposed method on CFG.
> >
> > **A.** Thank you for this important question. As discussed in the conclusion section, **our method is fully compatible with classifier-free guidance (CFG) or classifier guidance**, because our framework modifies how the vector field is trained, not how inference or conditioning mechanisms operate. If UOT-RFM is applied within the CFG framework, i.e.,
> > $$
> > \hat{v}\_{\theta} (t, x\_{t} | y) = (1-w) \cdot v\_{\theta} (t, x\_{t} | y = \varphi) + w \cdot v\_{\theta} (t, x\_{t} | y),
> > $$
> > the class-conditional flow matching model $v_{\theta} (t, x_{t} | y)$ would benefit from our reweighting scheme and better approximate minority or outlier modes within each class. In this regard, **UOT-RFM can potentially enhance CFG performance** by providing a more balanced and faithful baseline model, which in turn improves tail-class modeling under guided generation.
> >
> > However, CFG requires class labels (or conditioning signals) during training, whereas **our work focuses specifically on the label-free long-tailed setting**. Thus, CFG targets a different problem formulation and lies outside the primary scope of this paper. Nonetheless, we agree that studying how UOT-RFM interacts with CFG is an interesting direction for future work.

---

> > > ### Comment · Reviewer_msMD · 2025-11-28
> > >
> > > Thanks to the author`s rebuttal. I have carefully checked all the contents. The further clarification from the author successfully addresses the significant concerns that whether the FID gain comes from the diversity after alleviating the long-tailed problem. Meanwhile, the new ablation studies address my concerns about the insufficient ablation study. The minior concern is that this paper does not report the performance on ImageNet-LT (Considering the rebuttal time, i think this is the minor concern). Therefore, the major concerns have been addressed by author during the rebuttal. I will increase my score to marginal accept.

---

### Official Review · Reviewer_aJvH · 2025-10-26

**Soundness:** 2
**Presentation:** 3
**Contribution:** 2
**Rating:** 4
**Confidence:** 3

**Summary:**

This paper is about generative AI, in particular on image generation. The paper proposed an Unbalanced Optimal Transport method aimed at circumventing the problem of data imbalance during the training process. As imbalanced data may affect the training of a generative AI and in particular, introduce bias toward the long-tailed minority class, it is a critical problem in the field. The paper tries to address this problem via an inverse weighting strategy and tests show some improved results such as lower FID.

**Strengths:**

Strengths include the creation of UOTRFM method to reduce bias towards the long-tailed minority class, better FID in experiments, and the fact that the method does not induce too much extra computational time (7%).

**Weaknesses:**

The biggest weakness is that it requires the training to start from scratch to take into account class imbalance, as the paper acknowledges at the end, this may reduce the applicability of the method in real-world. The other weaknesses include that, while the proposed method had better FID, by visual inspection, quality of the generated images is not always better than those of comparison methods, though some images given by the proposed method look cleaner and less noisy, but it seems not all the generated images are visually better than those from other methods.
Another weakness is the selection of appropriate k seems ad-hoc, it is unclear why for a k>10, FID goes up in Figure 6(a).

**Questions:**

A better elaboration of the role or meaning of correction order k will be helpful. Is k related to the order of moments of the underlying distribution of the data?
What is the relationship between k and tau? As displayed in Figure 6(b) and (c), it seems there are some relationships between k and tau. And according to the paragraph starting at line 465, why a larger tau, which, according to the paper, giving better match for the UOT marginal density, does not lead to a smaller FID in the plots?

---

> ### Author Response · Authors · 2025-11-23
> **Response to Reviewer aJvH (1/2)**
>
> We sincerely thank the reviewer for carefully reading our manuscript and providing valuable feedback. We hope our responses to be helpful in addressing the reviewer's concerns. We highlighted the corresponding revisions in the manuscript in Blue.
>
>  $ $
>
> ---
> > [W1] The biggest weakness is that it requires the training to start from scratch to take into account class imbalance, as the paper acknowledges at the end, this may reduce the applicability of the method in real-world.
>
> **A.** We agree with the reviewer that the need for training from scratch is a limitation compared to test-time guidance methods. However, we believe that our approach still offers strong practical value. Our method aims to improve the **baseline (unguided) generative model** itself under long-tailed distributions, whereas test-time guidance techniques operate on a **fixed, pre-trained baseline** to extract more tail-class samples. Because these two approaches target different stages of the pipeline, they are not substitutes but rather complementary. A stronger baseline model produced by our method can serve as a better foundation for test-time guidance, enabling such guidance methods to achieve even better tail-class performance.
>
> $ $
>
> ---
> > [W2] The other weaknesses include that, while the proposed method had better FID, by visual inspection, quality of the generated images is not always better than those of comparison methods, though some images given by the proposed method look cleaner and less noisy, but it seems not all the generated images are visually better than those from other methods.
>
> **A.** We appreciate the reviewer’s careful observation regarding the visual quality of generated samples. As noted, while our method achieves significantly improved FID scores, which reflect overall marginal fidelity, it does not guarantee that every single generated image will look better than those from baselines. **We complement this limitation by incorporating more fine-grained evaluations from diverse perspectives**, such as Normalized Class Ratio Error (NCRE), precision, recall, and class-wise NLL. In particular, the class-wise NLL provides a more fine-grained evaluation of per-sample fidelity, addressing the fact that FID relies on a Gaussian approximation in feature space. **Across all metrics, our method consistently outperforms the baselines, indicating that the overall generation quality is improved** even if not every individual image is visually superior.
>
>
> $ $
>
> ---
> > [W3] Another weakness is the selection of appropriate k seems ad-hoc, it is unclear why for a k>10, FID goes up in Figure 6(a).
>
> **A.** As with many deep learning methods, the correction order $k$ in UOT-RFM works as a hyperparameter, and there is a “sweet spot” that leads to the best performance. In practice, as shown in Table 6, UOT-RFM   is quite robust within a reasonable range of $k$. For example, on CIFAR-10-LT, FID remains around 11 for $k \in \{6, 8, 10\}$ when $\tau = 2$.
>
> Moreover, our framework provides a principled interpretation of this parameter. The appropriate choice of $k$ depends on both the imbalance ratio of the long-tailed distribution (i.e., minority-to-majority sample count ratio) and the inherent majority bias in the flow matching framework. Hence, **when $k$ exceeds this sweet spot, the model begins to over-correct**, placing undesirably large weight on minority samples. As a result, the generated distribution
> $$
> p_{1} \propto s_{\tau}^{-(k-1)} \nu \quad \text{(Thm 3.1)}.
> $$
> becomes **overly skewed toward minority modes**. This over-emphasis explains the rise in FID for $k>10$, as shown in Fig. 6(a).
>
> $ $

---

> > ### Author Response · Authors · 2025-11-23
> > **Response to Reviewer aJvH (2/2)**
> >
> > ---
> > > [Q1] A better elaboration of the role or meaning of correction order k will be helpful. Is k related to the order of moments of the underlying distribution of the data? What is the relationship between k and tau? As displayed in Figure 6(b) and (c), it seems there are some relationships between k and tau.
> >
> > **A.** We thank the reviewer for the insightful question. **We clarified this relationship in Lines 466-470 of the revised manuscript**. As the reviewer suggested, the correction order $k$ and the marginal matching intensity  $\tau$ are closely related, and this relationship helps explain the patterns observed in Fig. 6(b).
> >
> > **The marginal matching intensity $\tau$ controls the variance of the majority score $s_{\tau}$**. In the Unbalanced Optimal Transport (UOT) formulation, $\tau=\tau_{2}$ determines how strongly the objective penalizes deviations of the target marginal $\pi_{1}$ from the true data distribution $\nu$.
> > $$
> >     C\_{UOT}(\mu, \nu) = \inf\_{\pi \in \mathcal{M}\_{+} (\mathcal{X}\times\mathcal{Y})} \biggl[ \int\_{\mathcal{X}\times \mathcal{Y}} c(x,y) d \pi(x,y)
> >     + \tau\_{1} D\_{\Psi\_1}(\pi\_0 \| \mu) + \tau\_{2} D\_{\Psi\_2}(\pi\_1 \| \nu) \biggr]
> > $$
> > Note that we fix $\tau_{1}=\infty$ to enforce $\pi_{0}=\mu$. When $\tau$ is large, the optimal marginal $\pi_1$ is forced to more closely match $\nu$, making the majority score $s_{\tau}(\cdot)$ closer to 1 for all samples. Conversely, when $\tau$ is small, $\pi_{1}$ is allowed to deviate more from $\nu$, amplifying the imbalance between majority and minority samples, which produces a wider spread in $s_{\tau}$.
> >
> > Therefore, when $\tau$ is small, even a modest correction order $k$ is sufficient to strongly adjust the generated distribution
> > $$
> > p_{1} \propto s_{\tau}^{-(k-1)} \nu \quad \text{(Thm. 3.1)}.
> > $$
> > This relationship is empirically validated in our ablation study on the correction order $k$ and the marginal matching intensity $\tau$ (Fig. 6(b)). **When $\tau$ is large, $s_{\tau} \approx 1$, so the model becomes less sensitive to the choice of $k$, resulting in a smaller variance in FID across different values of $k$**. In contrast, when $\tau$ is small, $s_{\tau}$ can have more extreme values. Hence, the FID score exhibit larger variation depending on $k$.
> >
> > $ $
> >
> > ---
> > > [Q2] And according to the paragraph starting at line 465, why a larger tau, which, according to the paper, giving better match for the UOT marginal density, does not lead to a smaller FID in the plots?
> >
> > **A.** This behavior arises from the **majority bias** inherent in the standard flow matching framework, which is precisely the motivation for introducing UOT-RFM. In principle, one might expect that using a very large $\tau$, which makes the UOT marginal $\pi_1$ closely match the data distribution $\nu$, would lead to better FID. **Under this perspective, OT-CFM (which corresponds to $\tau = \infty$) should achieve the best performance** because it enforces exact marginal matching.
> >
> > However, in practice, the flow matching shows the majority bias: head classes are well-modeled, but tail classes are badly generated or ignored. Simply forcing $\pi_1$ to match $\nu$ does not correct this bias during training. For this reason, we use **UOT with finite $\tau$ to extract a meaningful majority score of each sample without requiring any label information**. This score allows us to counteract majority bias via our correction term.

---

### Official Review · Reviewer_XNLv · 2025-11-01

**Soundness:** 2
**Presentation:** 2
**Contribution:** 2
**Rating:** 2
**Confidence:** 2

**Summary:**

This paper introduces UOT-Reweighted Flow Matching (UOT-RFM), a framework for improving generative modeling under class imbalance without using labels. The general motivation is that standard flow matching suffers from majority bias, producing low-quality or underrepresented samples for minority modes. To address this, the authors build on Unbalanced Optimal Transport (UOT) to compute couplings between mini-batches and define a majority score, a label-free measure derived from the density ratio between the UOT target marginal and the data distribution. During training, the flow matching loss is reweighted by the inverse of this score raised to a power k, which the authors claim corrects majority bias and enhances tail generation. The paper reports improvements on long-tailed CIFAR-10-LT and CIFAR-100-LT, where UOT-RFM yields lower FID and higher recall than OT- or independent-coupling baselines. The method performs comparably on balanced datasets. While conceptually interesting, the empirical validation is limited to small-scale experiments and does not rigorously test the estimator or the theoretical assumptions that motivate the reweighting.

**Strengths:**

### **Strengths**

- The paper identifies a concrete and relevant limitation of flow matching, namely its bias toward majority modes when trained on long-tailed data, and proposes a simple label-free correction mechanism based on reweighting.
- The method integrates cleanly with existing flow matching pipelines and is easy to implement, which makes it accessible for future work.
- The idea of computing a majority score through unbalanced optimal transport couplings is conceptually neat and connects geometric properties of the data with reweighting in an unsupervised way.
- The experimental results, though small in scale, show promising trends, including improved recall and better alignment with class proportions, while maintaining reasonable computational efficiency.

**Weaknesses:**

### **Weaknesses**

- The experimental scope seems too limited for a venue like ICLR. All evaluations are performed on CIFAR-10-LT and CIFAR-100-LT at 32×32 resolution, with proxy labels from a pretrained classifier. There is no evidence that the method scales to higher-resolution datasets or other modalities. The absence of larger-scale tests or statistical confidence intervals makes the empirical validation insufficient to justify the claims. This is my main critique of the work in its current form.
- Perhaps I do not understand this point well, but it seems the estimator for the majority score is not examined very well. Specifically, there is no analysis of whether it truly captures head versus tail dominance. Since this estimator underlies the central idea of the paper, its unverified behavior weakens the overall contribution.
- The recall-precision trade-off observed in the results is not analyzed. Precision consistently drops when recall improves, but the paper does not investigate which classes are affected or whether this change is visually or statistically acceptable.

**Questions:**

### **Questions**

-Can the authors provide empirical evidence that the majority-score estimator correlates with head and tail dominance, for example through visualization, correlation statistics, or controlled toy experiments?
- How does the method perform on larger and more realistic datasets, such as ImageNet-LT or Places-LT, and can the authors report multiple seeds and confidence intervals for FID, precision, and recall?
- Would using a feature-space distance rather than pixel-space distance in the UOT computation improve the majority-score quality and model performance?
- How sensitive is the approach to the parameters? Specifically, does the over-correction regime that improves recall also consistently degrade precision, and if so, can it be balanced systematically?

---

> ### Author Response · Authors · 2025-11-23
> **Response to Reviewer XNLv (1/2)**
>
> We sincerely thank the reviewer for carefully reading our manuscript and providing valuable feedback. We hope our responses to be helpful in addressing the reviewer's concerns. We highlighted the corresponding revisions in the manuscript in Red.
>
> $ $
>
> ---
> > [W1] The experimental scope seems too limited for a venue like ICLR. All evaluations are performed on CIFAR-10-LT and CIFAR-100-LT at 32×32 resolution, with proxy labels from a pretrained classifier. There is no evidence that the method scales to higher-resolution datasets or other modalities. The absence of larger-scale tests or statistical confidence intervals makes the empirical validation insufficient to justify the claims. This is my main critique of the work in its current form.
>
> > [Q2] How does the method perform on larger and more realistic datasets, such as ImageNet-LT or Places-LT, and can the authors report multiple seeds and confidence intervals for FID, precision, and recall?
>
> **A.** We appreciate the reviewer for the valuable advice. Following the reviewer's suggestions, we conducted additional experiments addressing (1) statistical confidence intervals and (2) higher-resolution datasets.
>
> (1) We measured the statistical confidence intervals by running two additional runs for our main evaluation setup, i.e., CIFAR-10-LT with an imbalance ratio $\mathcal{I} = 0.01$ and by reporting mean $\pm$ standard deviation. The results are as follows:
>
> | Model | CIFAR-10-LT (FID)|
> |:--- |:---|
> |OT-CFM| 17.34 $\pm$ 0.23 |
> |Ours (UOT-RFM)| 11.17 $\pm$ 0.15 |
>
> Although we have not yet completed additional multi-seed runs for all baselines due to time constraints, the variance of UOT-RFM is small ($\sigma = 0.15$) and still yields a large and statistically meaningful gap over other baselines such as I-CFM (14.57) and UOT-CFM (14.25). The other baseline results will be reported.
>
> (2) Due to time and resource constraints during the rebuttal period, we further applied our methods to FFHQ-64 under the latent flow matching framework. Although we were able to test only a limited set of hyperparameter settings, UOT-RFM already shows promising improvements over OT-CFM  (under latent flow matching):
>
> | Model | FFHQ-64 (FID)|
> |:--- |:---|
> |OT-CFM| 14.24  |
> |Ours (UOT-RFM)| 13.53 |
>
> Here, UOT-RFM was tested with hyperparameters $\tau=4$ and $k=4$. These results demonstrate that UOT-RFM scales beyond 32×32 resolution and continues to outperform strong OT-based baselines. We are conducting further hyperparameter tuning and will report updated results during the discussion phase if available.
>
> $ $
>
> ---
> > [W2] Perhaps I do not understand this point well, but it seems the estimator for the majority score is not examined very well. Specifically, there is no analysis of whether it truly captures head versus tail dominance. Since this estimator underlies the central idea of the paper, its unverified behavior weakens the overall contribution.
>
> > [Q1] Can the authors provide empirical evidence that the majority-score estimator correlates with head and tail dominance, for example through visualization, correlation statistics, or controlled toy experiments?
>
> **A.** We appreciate the reviewer for the thoughtful comment. To directly evaluate whether the majority-score estimator reflects head–tail dominance, we conducted an additional synthetic experiment.
>
> Experimental settings are as follows: We constructed a **3-component 2D Gaussian mixture** as the *target distribution, with clearly imbalanced mixture weights* (Head: 0.70, medium-tail: 0.275, extreme tail: 0.025). *For the source distribution, we used a single-mode Gaussian*. The majority score is computed using the unbalanced Sinkhorn solver. Note that this is identical to how UOT-RFM estimates the majority score between mini-batches.
>
> The experimental results are included in Fig. 14 and Table 5 in Appendx F3. This result shows that **our majority score is correlated with the head-tail dominance**. The qualitative score visualization and the average score for each component (1.21: 0.53: 0.5 for head: medium-tail: tail at $\tau =5$) show a clear ordering:
> $$
> \text{head} > \text{medium-tail} > \text{extreme-tail}.
> $$
> This ordering exactly matches the true mixture weights. Therefore, this experiment gives direct evidence that our majority score estimator can correctly capture head-tail dominance.

---

> > ### Author Response · Authors · 2025-11-23
> > **Response to Reviewer XNLv (2/2)**
> >
> > ---
> > > [W3] The recall-precision trade-off observed in the results is not analyzed. Precision consistently drops when recall improves, but the paper does not investigate which classes are affected or whether this change is visually or statistically acceptable.
> >
> > **A.** We appreciate the reviewer’s comment regarding the recall–precision trade-off. As shown in Table 3 (CIFAR-10-LT and CIFAR-100-LT), our method improves recall while exhibiting a slightly lower precision. **To assess this trade-off from a more holistic perspective, we conducted a comprehensive analysis using diverse evaluation metrics**:
> > - F1-score (Table 3): captures the overall balance between recall and precision (e.g., 0.42 $\to$ 0.49).
> > - FID (Table 1): evaluates marginal fidelity (e.g., 14.57 $\to$ 11.03).
> > - Generated class distribution (Figure 4 and NCRE metric, Lines 404–407): assesses the fidelity of class ratios (e.g., 0.84 $\to$ 0.40).
> > - Class-wise NLL (Figures 11 and 13): examines fidelity at the per-class level.
> >
> > We would like to emphasize that our UOT-RFM consistently achieves substantial improvements over other flow matching baselines. In particular, the **class-wise NLL improvements, indicating the class-wise fidelity, are observed across almost all classes** (Figs. 11 and 13). We believe this result addresses the concern regarding the class-wise evaluation.
> >
> > $ $
> >
> > ---
> > > [Q3] Would using a feature-space distance rather than pixel-space distance in the UOT computation improve the majority-score quality and model performance?
> >
> > **A.** Thank you for the insightful question. We agree that using a feature-space distance (e.g., in a pretrained encoder) could potentially yield a majority score that better reflects semantic structure. In this work, however, our goal was to first establish that **UOT-RFM is feasible and effective even when operating entirely in raw pixel space**, without relying on auxiliary feature extractors.
> >
> > Following the reviewer’s suggestion (and as discussed in [W1]), we extended our evaluation to FFHQ-64 under the **latent flow matching** framework, where distances are computed in a learned feature space. Although we were able to test only a limited set of hyperparameter settings, **UOT-RFM already shows promising improvements over OT-CFM** in this setting.  (See [W1] for the corresponding result table.) This result indicates that our approach can extend to feature-space formulations as well.
> >
> > $ $
> >
> > ---
> > > [Q4] How sensitive is the approach to the parameters? Specifically, does the over-correction regime that improves recall also consistently degrade precision, and if so, can it be balanced systematically?
> >
> > **A.** Thank you for the insightful question. **The marginal matching intensity  $\tau$ and the correction order $k$ are closely related**. The marginal matching intensity $\tau$ controls the variance of the majority score $s_{\tau}$. Based on the Unbalanced OT formulation, when $\tau$ is large, the majority score $s_{\tau}(\cdot)$ is closer to 1 for all samples. Conversely, when $\tau$ is small, $s_{\tau}$ exhibits a wide spread. Therefore, when $\tau$ is small, even a modest correction order $k$ is sufficient to strongly adjust the generated distribution
> > $$
> > p_{1} \propto s_{\tau}^{-(k-1)} \nu \quad \text{(Thm. 3.1)}.
> > $$
> >
> > As discussed in W2, we conducted a synthetic experiment to verify this relationship between $\tau$ and $s_{\tau}$ (Table 5 in Appendx F3). We constructed a 3-component 2D Gaussian mixture as the *target distribution, with clearly imbalanced mixture weights* (Head: 0.70, medium-tail: 0.275, extreme tail: 0.025). *For the source distribution, we used a single-mode Gaussian*. **Average majority scores for each component behave monotonically with respect to $\tau$**:
> >
> > |$\tau$| head| medium-tail | extreme-tail |
> > |:--- |:---|:---|:---|
> > |$\tau =1$| 1.43 | 0.05 |0.04 |
> > |$\tau =3$| 1.31 |0.33 |0.33 |
> > |$\tau =5$| 1.21 |0.53 |0.50 |
> > |$\tau =10$| 1.12 |0.73 |0.72 |
> >
> > These results confirm that increasing $\tau$ compresses the spread of $s_\tau$, consistent with the theoretical interpretation.
> >
> > Moreover, the relationship between $\tau$ and $k$ is **empirically validated in our ablation study on the correction order $k$ (Fig. 6(b) and Tables 9 and 10)**. As shown in Fig. 6(b), the variance of FID scores across different values of $k$ is larger when $\tau$ is small. Moreover, as the reviewer predicted, increasing $k$ consistently improves recall while gradually degrading precision until reaching a “sweet spot.” Therefore, in practice, **this trade-off can be systematically balanced by jointly tuning $\tau$ and $k$**: a smaller $\tau$ requires only a small $k$ for effective correction, while a larger $\tau$ may require a larger $k$ but at greater risk of precision degradation.

---

> > ### Author Response · Authors · 2025-11-25
> > **Response to Reviewer XNLv (1-1/2)**
> >
> > Dear Reviewer XNLv,
> >
> > We also obtained **preliminary results on the Places-LT dataset (256$\times$256)** under the latent flow matching framework. For this initial test, we simply reused the best hyperparameters from CIFAR-10-LT ($\tau = 2$, $k = 10$) **without any dataset-specific tuning**. Nevertheless, **UOT-RFM outperforms OT-CFM**, demonstrating that our method remains effective on higher-resolution, large-scale long-tailed datasets:
> >
> > | Model | Places-LT-256 (FID $\downarrow$ )|
> > |:--- |:---|
> > |OT-CFM| 18.7  |
> > |Ours (UOT-RFM)| 18.1 |
> >
> > These preliminary results support the scalability and feasibility of UOT-RFM beyond 32×32 settings, addressing the reviewer’s concern regarding performance on more realistic datasets. Further hyperparameter exploration is ongoing, and we will report any improved results during the discussion phase.

---

### Meta-Review · Area_Chair_DYsu · 2026-01-02

**Summary:**

The paper introduces a re-weighting strategy for flow matching for label-free long-tailed generation (i.e. generation with unbalanced classes) based on Unbalanced Optimal Transport (OT). The main contribution lies in leveraging unbalanced OT theory to estimate the degree of majority directly without the need for class labels. These estimates are then used as weights to balance the data distribution during training. Experimental results show that the proposed method effectively mitigates the long-tailed problem.

**Reviewer Concerns:**

A liested weakness is that the  training has to be restarted from scratch to take into account class imbalance. Authors agree but arguee that their method can be used complementary to test-time guided methods.
Another argument was that FID improves but visual quality not. In retun the authors incorporated additional evaluation metrices that show the superiority of the appeoach.
Moroever, reviweres wanted different solvers to be tested and authors showed that Euler solver leads so slighly worse results,
Also authors added an ablation study showing principled parameter selection. Therefore, a lot of concerns were resolved in the rebuttal.

Two main concerns are not fully resolved from my perspective. In the orignal paper, there was no evaluation of confidence intervalls.  Authors performed 2 additional runs for their method and worst baseline, showing a relative small variance. The variance in the results for sronger baselines was however not analyzed so far and for proper statistical significance test a minimum of 5 runs would be needed.
Also in respons to the reviews authors added results on data sets with higher resultion that showed a smaller gap in FID values, and here variance is not measured. Therefore, the statistical significance for larger scale experiements is not totally clear and the paper would highly benefit from adding this additional experiments.

**Reviewer Scores:**

Reviewer msMD indicated that they will increase thir score from 4 to 6.
The other reviewers did not respond before the occurance of the leak but I would estimate that maybe aJvH would have changed from 4 to 6 as well. Reviewer  XNLv might also have inceased their score (he indicated low confidence in the primary review), but overall I daught that they would increas to 6. Therefore, overall the paper is boarderline and given the missing experiments showing statistical significance of the results on higher resolution data sets,  I think the paper should not be accepted.

---

### Decision · Program_Chairs · 2026-01-26

Reject